# Interpreting Attention Layer Outputs with Sparse Autoencoders

## Abstract

Decomposing model activations into interpretable components is a key open problem in mechanistic interpretability. Sparse autoencoders (SAEs) are a popular method for decomposing the internal activations of trained transformers into sparse, interpretable features, and have been applied to MLP layers and the residual stream. In this work we train SAEs on attention layer outputs and show that also here SAEs find a sparse, interpretable decomposition. We demonstrate this on transformers from several model families and up to 2B parameters. We perform a qualitative study of the features computed by attention layers, and find multiple families: long-range context, short-range context and induction features. We qualitatively study the role of every head in GPT-2 Small, and estimate that at least 90% of the heads are polysemantic, i.e. have multiple unrelated roles. Further, we show that Sparse Autoencoders are a useful tool that enable researchers to explain model behavior in greater detail than prior work. For example, we explore the mystery of why models have so many seemingly redundant induction heads, use SAEs to motivate the hypothesis that some are long-prefix whereas others are short-prefix, and confirm this with more rigorous analysis. We use our SAEs to analyze the computation performed by the Indirect Object Identification circuit (Wang et al. (2023)), validating that the SAEs find causally meaningful intermediate variables, and deepening our understanding of the semantics of the circuit. We open-source the trained SAEs and a tool for exploring arbitrary prompts through the lens of Attention Output SAEs.

## 1 Introduction

Mechanistic interpretability aims to reverse engineer neural network computations into human-understandable algorithms (Olah, 2022; Olah et al., 2020). A key sub-problem is to decompose high dimensional activations into meaningful concepts, or features. If successful at scale, this research would enable us to identify and debug model errors (Hernandez et al., 2022; Vig et al., 2020; Gandelsman et al., 2024; Marks et al., 2024), control and steer model behavior (Panickssery et al., 2024; Turner et al., 2023; Zou et al., 2023), and better predict out-of-distribution behavior (Mu & Andreas, 2020; Carter et al., 2019; Goh et al., 2021).

Prior work has successfully analyzed many individual model components, such as neurons and attention heads. However, both neurons (Wang et al., 2023) and attention heads (Gould et al., 2023) are often *polysemantic* (Olah et al., 2017): they appear to represent multiple unrelated concepts or perform different functions depending on the input. Polysemanticity makes it challenging to interpret the role of individual neurons or attention heads in the model's overall computation, suggesting the need for alternative units of analysis.

Our paper builds on literature using Sparse Autoencoders (SAEs) to extract interpretable feature dictionaries from the residual stream (Cunningham et al., 2023; Yun et al., 2023) and MLP activations (Bricken et al., 2023). While these approaches have shown promise in disentangling activations into interpretable features, attention layers have remained difficult to interpret. In this work, we apply SAEs to reconstruct attention layer outputs, and develop a novel technique (**weight based head attribution**) to associate learned features with specific attention heads. This allows us to sidestep challenges posed by polysemanticity (Section 2).

| Feature: | Induction | Local context | Succession |
|---|---|---|---|
| Description: | "I previously preceded board" | "I am in a question" | "16 came before, add 1" |
| Feature Activations: | blackboard: shared **black** | 3. **Which pass option** | suspects, aged 16 **to** |
| DFA: | black**board**: shared black | 3. **Which** pass option | suspects, aged **16** to |
| Top Logit: | "board" | "?" | "17" |

Figure 1: Overview. We train Sparse Autoencoders (SAEs) on $\mathbf{z}_{cat}$, the attention layer outputs pre-linear, concatenated across all heads. The SAEs extract linear directions that correspond to concepts in the model, giving us insight into what attention layers learn in practice. Further, we uncover what information was used to compute these features with direct feature attribution (DFA, Section 2).

Since SAEs applied to LLM activations are already widely used in the field, we do not see the application of SAEs to attention outputs as our main contribution. Instead, we hope our main contribution to be making a case for Attention Output SAEs as a valuable research tool that others in the mechanistic interpretability community should adopt. We do this by rigorously showing that Attention Output SAEs find sparse, interpretable reconstructions, that they easily enable qualitative analyses to gain insight into the functioning of attention layers, and that they are a valuable tool for novel research questions such as why models have so many seemingly redundant induction heads (Olsson et al., 2022) or better understanding the semantics of the Indirect Object Identification circuit (Wang et al., 2023).

In more detail, our main contributions are as follows:

1. We demonstrate that Sparse Autoencoders decompose attention layer outputs into sparse, interpretable linear combinations of feature vectors, giving us deeper insight into what concepts attention layers learn up to 2B parameter models (Section 3). We perform a qualitative study of the features computed by attention layers, and find multiple families: long-range context, short-range context and induction features (Section 3.3).
2. We apply SAEs to systematically inspect every attention head in GPT-2 Small (Section 4.1), and extend this analysis to make progress on the open question of why there are multiple, seemingly redundant induction heads (Section 4.2). Our method identifies differences between induction heads (Olsson et al., 2022) which specialize in "long prefix induction" (Goldowsky-Dill et al., 2023) vs "short prefix induction", demonstrating the utility of these SAEs for interpretability research.
3. We show that Attention Output SAEs are useful for circuit analysis (Section 4.3), by finding and interpreting causally relevant SAE features for the widely-studied Indirect Object Identification circuit (Wang et al., 2023), and resolving a way our prior understanding was incomplete.

## 2 METHODOLOGY

**Reconstructing attention layer outputs:** We closely follow the setup from Bricken et al. (2023) to train Sparse Autoencoders that reconstruct the attention layer outputs. Specifically, we train our SAEs on the $\mathbf{z} \in \mathbb{R}^{d_{\text{head}}}$ vectors (Nanda & Bloom, 2022) concatenated across all heads of some arbitrary layer (i.e. $\mathbf{z}_{\text{cat}} \in \mathbb{R}^{d_{\text{model}}}$ where $d_{\text{model}} = n_{\text{heads}} \cdot d_{\text{head}}$). Note that $\mathbf{z}$ is the attention weighted sum of value vectors $\mathbf{v} \in \mathbb{R}^{d_{\text{head}}}$ *before* they are converted to the attention output by a linear map (Figure 1), and should not be confused with the final output of the attention layer. We choose to concatenate each $\mathbf{z}$ vector in the layer, rather than training an SAE per head, so that our method is robust to features represented as a linear combination of multiple head outputs (Olah et al., 2023).

Given an input activation $\mathbf{z}_{\text{cat}} \in \mathbb{R}^{d_{\text{model}}}$, Attention Output SAEs compute a decomposition (using notation similar to Marks et al. (2024)):

$$\mathbf{z}_{\text{cat}} = \hat{\mathbf{z}}_{\text{cat}} + \varepsilon(\mathbf{z}_{\text{cat}}) = \sum_{i=0}^{d_{\text{sae}}} f_i(\mathbf{z}_{\text{cat}})\mathbf{d}_i + \mathbf{b} + \varepsilon(\mathbf{z}_{\text{cat}}) \tag{1}$$

where $\hat{\mathbf{z}}_{\text{cat}}$ is an approximate reconstruction and $\varepsilon(\mathbf{z}_{\text{cat}})$ is an error term. We define $\mathbf{d}_i$ as unit-norm *feature directions* with sparse coefficients $f_i(\mathbf{z}_{\text{cat}}) \geq 0$ as the corresponding *feature activations* for $\mathbf{z}_{\text{cat}}$. We also include an SAE bias term $\mathbf{b}$.

As mentioned, we do not train SAEs on the output of the attention layer $W_O\mathbf{z}_{\text{cat}} \in \mathbb{R}^{d_{\text{model}}}$ (where $W_O$ is the out projection weight matrix of the attention layer (Figure 1)). Since $W_O\mathbf{z}_{\text{cat}}$ is a linear transformation of $\mathbf{z}_{\text{cat}}$, we expect to find similar features. However, we deliberately trained our SAE on $\mathbf{z}_{\text{cat}}$ since we find that this allows us to attribute which heads the decoder weights are from for each SAE feature, as described below.

**Weight-based head attribution:** We develop a technique specific to this setup: decoder weight attribution by head. For each layer, our attention SAEs are trained to reconstruct $\mathbf{z}_{\text{cat}}$, the concatenated outputs of each head. Thus each SAE feature direction $\mathbf{d}_i$ is a vector in $\mathbb{R}^{n_{\text{heads}} \cdot d_{\text{head}}}$.

We can split each feature direction, $\mathbf{d}_i$, into a concatenation of $n_{\text{heads}}$ smaller vectors, each of shape $d_{\text{head}}$: $\mathbf{d}_i = [\mathbf{d}_{i,1}^\top, \mathbf{d}_{i,2}^\top, \ldots, \mathbf{d}_{i,n_{\text{heads}}}^\top]^\top$ where $\mathbf{d}_{i,j} \in \mathbb{R}^{d_{\text{head}}}$ for $j = 1, 2, \ldots, n_{\text{heads}}$.

We can intuitively think of each $\mathbf{d}_{i,j}$ as reconstructing the part of feature direction that comes from head $j$. We then compute the norm of each slice as a proxy for how strongly each head writes this feature. Concretely, for any feature $i$, we can compute the weights based attribution score to head $k$ as

$$h_{i,k} = \frac{\|\mathbf{d}_{i,k}\|_2}{\sum_{j=1}^{n_{\text{heads}}} \|\mathbf{d}_{i,j}\|_2} \tag{2}$$

For any head $k$, we can also sort all *features* by their head attribution to get a sense of what features that head is most responsible for outputting (see Section 4.1).

**Direct feature attribution:** We provide an activation based attribution method to complement the weights based attribution above. As attention layer outputs are a linear function of attention head outputs (Elhage et al., 2021), we can rewrite SAE feature activations in terms of the contribution from each head.

$$f_i^{\text{pre}}(\mathbf{z}_{\text{cat}}) = \mathbf{w}_i^\top \mathbf{z}_{\text{cat}} = \mathbf{w}_{i,1}^\top \mathbf{z}_1 + \mathbf{w}_{i,2}^\top \mathbf{z}_2 + \cdots + \mathbf{w}_{i,n_{\text{heads}}}^\top \mathbf{z}_{n_{\text{heads}}} \tag{3}$$

where $\mathbf{w}_i \in \mathbb{R}^{d_{\text{model}}}$ is the ith row of the encoder weight matrix, $\mathbf{w}_{i,j} \in \mathbb{R}^{d_{\text{head}}}$ is the jth slice of $\mathbf{w}_i$, and $f_i^{\text{pre}}(\mathbf{z}_{\text{cat}})$ is the pre-ReLU feature activation for feature $i$ (i.e. $\text{ReLU}(f_i^{\text{pre}}(\mathbf{z}_{\text{cat}})) := f_i(\mathbf{z}_{\text{cat}})$). Note that we exclude SAE bias terms for brevity.

We call this "direct feature attribution" (as it's analogous to direct logit attribution (Nanda, 2022c)), or "DFA" by head. We apply the same idea to perform direct feature attribution on the value vectors

Table 1: Evaluations of sparsity, fidelity, and interpretability for Attention Output SAEs trained across multiple models and layers. Percentage of interpretable features were based on 30 randomly sampled live features inspected per layer.

| Model | Layer | L0 | % CE Rec.† | % Interp. | Width |
|---|---|---|---|---|---|
| Gemma-2B (Gemma Team et al., 2024) | 6 | 90 | 75% | 66% | 16384 |
| GPT-2 Small | 0 | 3 | 99% | 97% | 24576 |
| GPT-2 Small | 1 | 20 | 78% | 87% | 24576 |
| GPT-2 Small | 2 | 16 | 90% | 97% | 24576 |
| GPT-2 Small | 3 | 15 | 84% | 77% | 24576 |
| GPT-2 Small | 4 | 15 | 88% | 97% | 24576 |
| GPT-2 Small | 5 | 20 | 85% | 80% | 49152 |
| GPT-2 Small | 6 | 19 | 82% | 77% | 24576 |
| GPT-2 Small | 7 | 19 | 83% | 70% | 49152 |
| GPT-2 Small | 8 | 20 | 76% | 60% | 24576 |
| GPT-2 Small | 9 | 21 | 83% | 77% | 24576 |
| GPT-2 Small | 10 | 16 | 85% | 80% | 24576 |
| GPT-2 Small | 11 | 8 | 89% | 63% | 24576 |
| GPT-2 Small | All | | | 80% | |
| GELU-2L (Nanda, 2022b) | 1 | 12 | 87% | 83% | 16384 |

† Percentage of cross-entropy loss recovered (Equation 4).

at each source position, since the $\mathbf{z}$ vectors are a linear function of the value vectors if we freeze attention patterns (Elhage et al., 2021; Chughtai et al., 2024). We call this "DFA by source position".

# 3 ATTENTION OUTPUT SAES FIND SPARSE, INTERPRETABLE RECONSTRUCTIONS

In this section, we show that Attention Output SAE reconstructions are sparse, faithful, and interpretable. We first explain the metrics we use to evaluate our SAEs (Section 3.1). We then show that our SAEs find sparse, faithful, interpretable reconstructions (Section 3.2). Finally we demonstrate that our SAEs give us better insights into the concepts that attention layers learn in practice by discovering three attention feature families (Section 3.3).

## 3.1 SETUP

To evaluate the sparsity and fidelity of our trained SAEs we use two metrics from Bricken et al. (2023) (using notation similar to Rajamanoharan et al. (2024)):

**L0.** The average number of features firing on a given input, i.e. $\mathbb{E}_{\mathbf{x} \sim D} \|\mathbf{f}(\mathbf{x})\|_0$.

**Loss recovered.** The average cross entropy loss of the language model recovered with the SAE "spliced in" to the forward pass, relative to a zero ablation baseline. More concretely:

$$1 - \frac{\text{CE}(\hat{\mathbf{x}} \circ \mathbf{f}) - \text{CE}(\text{Id})}{\text{CE}(\zeta) - \text{CE}(\text{Id})}, \tag{4}$$

where $\hat{\mathbf{x}} \circ \mathbf{f}$ is the autoencoder function, $\zeta : \mathbf{x} \to \mathbf{0}$ is the zero ablation function and Id: $\mathbf{x} \to \mathbf{x}$ is the identity function. According to this definition, an SAE that reconstructs its inputs perfectly would get a loss recovered of 100%, whereas an SAE that always outputs the zero vector as its reconstruction would get a loss recovered of 0%.

**Feature Interpretability Methodology.** We use dashboards (Bricken et al., 2023; McDougall, 2024) showing which dataset examples SAE features maximally activate on to determine whether

they are interpretable. These dashboards also show the top Direct Feature Attribution by source position, weight-based head attribution for each head (Section 2), approximate direct logit effects (Bricken et al., 2023) as well as activating examples from randomly sampled activation ranges, giving a holistic picture of the role of the feature. See Appendix C for full details about this methodology.

## 3.2 EVALUATING ATTENTION OUTPUT SAEs

We train and evaluate Attention Output SAEs across a variety of different models and layers. For GPT-2 Small (Radford et al., 2019), we notably evaluate an SAE for every layer. We find that our SAEs are sparse (oftentimes with $< 20$ average features firing), faithful (oftentimes $> 80\%$ of cross entropy loss recovered relative to zero ablation) and interpretable (oftentimes $> 80\%$ of live features interpretable). See Table 1 for per model and layer details.[1] Further detail regarding SAE training, such as the hyperparameters and data used, are provided in Appendix B. We also provide further discussion of the evaluation metrics and comparison to SAEs from prior work in Appendix I.

## 3.3 EXPLORING FEATURE FAMILIES

In this section we more qualitatively show that Attention Output SAEs are interpretable by examining different feature families: groups of SAE features that share some common high-level characteristic.

We first evaluate 30 randomly sampled live features from SAEs across multiple models and layers (as described in Section 3.1) and report the percentage of features that are interpretable in Table 1. We notice that in all cases, the majority of live features are interpretable, often >80%. Note that this is a small sample of features, and human judgment may be flawed. We list confidence intervals for percentage of interpretable features in Appendix C.1.

We now use our understanding of these extracted features to share deeper insights into the concepts attention layers learn. Attention Output SAEs enable us to taxonomize a large fraction of what these layers are doing based on feature families, giving us better intuitions about how transformers use attention layers in practice. Throughout our SAEs trained on multiple models, we repeatedly find three common feature families: induction features (e.g. "board" token is next by induction), local context features (e.g. current sentence is a question, Figure 1), and high-level context features (e.g. current text is about pets). All of these features involve moving prior information with the context, consistent with the high-level conceptualization of the attention mechanism from Elhage et al. (2021). We present these for illustrative purposes and do not expect these to nearly constitute a complete set of feature families. We also share additional feature families discovered in GPT-2 Small in Appendix P.

To more rigorously understand these feature families, we performed a case study for each of these features (similar to Bricken et al. (2023)). For brevity, we highlight a case study of an induction feature below and leave the remaining to Appendix G and H.

**Induction features.** Our analysis revealed multiple "induction features" across different models studied. As we are not aware of any induction features extracted by MLP SAEs in prior work, we hypothesize that induction features are uniquely computed by the attention layers (Bricken et al., 2023). In what follows, we showcase a "'board' is next by induction" feature from our L1 GELU-2L (Nanda, 2022b) SAE. However, we note that "board induction" is just one example from hundreds of "<token> is next by induction" features discovered by our analysis (see Appendix J). We also detail the feature's upstream computations and downstream effects in Appendix F.

The 'board' induction feature activates on the second instance of <token> in prompts of the form "<token> board . . . <token>". To demonstrate 'board induction' is a genuinely *monosemantic* feature, we provide evidence that the feature is both: (i) specific and (ii) sensitive to this context (Bricken et al., 2023).

---

[1]We release weights for every SAE, corresponding feature dashboards, and an interactive tool for exploring several attention SAEs throughout a model in Appendix A.

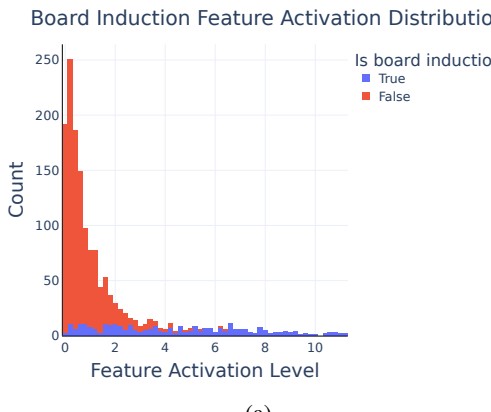

Figure 2: Specificity plot (Bricken et al., 2023) (a) which compares the distribution of the board induction feature activations to the activation of our proxy. The expected value plot (b) shows distribution of feature activations weighted by activation level (Bricken et al., 2023), compared to the activation of the proxy. Note red is stacked on top of blue, where blue represents examples that our proxy identified as board induction. We notice high specificity above the weakest feature activations.

Specificity was established through creation of a proxy that checks for cases of 'board' induction. Thereafter, we compared the activation of our proxy to the activation of the feature. We found that the upper parts of the activation spectrum clearly responded, with high specificity, to 'board' induction (Figure 2). Although some false positives were observed in the lower activation ranges (as in Bricken et al. (2023)), we believe there are mundane reasons to expect such results (see Appendix E).

We now move onto sensitivity. Our activation sensitivity analysis found 68 false negatives in a dataset of 1 million tokens, and all false negatives were manually checked. Although these examples satisfy the 'board' induction pattern, it is clear that 'board' should not be predicted. Often, this was because there were even stronger cases of induction for another token (Appendix D).

## 4 INTERPRETABILITY INVESTIGATIONS USING ATTENTION OUTPUT SAEs

In this section we demonstrate that Attention Output SAEs are useful as general purpose interpretability tools, allowing for novel insights about the role of attention layers in language models. We first develop a technique that allows us to systematically interpret every attention head in a model (Section 4.1), discovering new behaviors and gaining high-level insight into the phenomena of attention head polysemanticity (Janiak et al., 2023; Gould et al., 2023). We then apply our SAEs to make progress on the open question of why models have many seemingly redundant induction heads (Olsson et al., 2022), finding induction heads with subtly different behaviors: some primarily perform induction where there is a long prefix (Goldowsky-Dill et al., 2023) whereas others generally perform short prefix induction (Section 4.2). Finally, we apply Attention Output SAEs to circuit analysis (Section 4.3), unveiling novel insights about the Indirect Object Identification circuit (Wang et al., 2023) that were previously out-of-reach, and find causally relevant SAE features in the process.

### 4.1 INTERPRETING ALL HEADS IN GPT-2 SMALL

In this section, we use our weight-based head attribution technique (see Section 2) to systematically interpret every attention head in GPT-2 Small (Radford et al., 2019). As in Section 2, we apply Equation 2 to compute the weights based attribution score $h_{i,k}$ to each head $k$ and identify the top ten features $\{\mathbf{d}_{i_r}\}_{r=1}^{10}$ with highest attribution score to head $k$. Although Attention Output SAE features are defined relative to an entire attention layer, this identifies the features most salient to a given head with minimal contributions from other heads.

Using the feature interpretability methodology from Section 3.1, we manually inspect these features for all 144 attention heads in GPT-2 Small. Broadly, we observe that features become more abstract in middle-layer heads and then taper off in abstraction at late layers:

**Early heads.** *Layers 0-3* exhibit primarily syntactic features (single-token features, bigram features) and fire secondarily on specific verbs and entity fragments. Some long and short range context tracking features are also present.

**Middle heads.** *Layers 4-9* express increasingly more complex concept feature groups spanning grammatical and semantic constructs. Examples include heads that express primarily families of related active verbs, prescriptive and active assertions, and some entity characterizations. Late-middle heads show feature groups on grammatical compound phrases and specific concepts, such as reasoning and justification related phrases and time and distance relationships.

**Late heads.** *Layers 10-11* continue to express some complex concepts such as counterfactual and timing/tense assertions, with the last layer primarily exhibiting syntactic features for grammatical adjustments and some bigram completions.

We identify many existing known motifs (including induction heads (Olsson et al., 2022; Kissane, 2023), previous token heads (Voita et al., 2019; Kissane, 2023), successor heads (Gould et al., 2023) and duplicate token heads (Wang et al., 2023; Kissane, 2023)) in addition to new motifs (e.g. preposition mover heads). More details on each layer and head are available in Appendix S. Our method also suggests that over 90% of attention heads in GPT-2 Small are polysemantic, as we find features corresponding to multiple unrelated tasks. In Appendix K we provide more detail, including an example polysemantic head in GPT-2 Small. We note that there are some limitations to this methodology, as discussed in Appendix S.1.

## 4.2 LONG-PREFIX INDUCTION HEAD

In this section we apply Attention Output SAEs to make progress on a long-standing open question: why do models have so many seemingly redundant induction heads (Olsson et al., 2022)? We use our weight-based head attribution technique (see Section 4.1) to inspect the top SAE features attributed to two different induction heads and find one which specializes in "long prefix induction" (Goldowsky-Dill et al., 2023), while the other primarily does "short prefix induction".

As a case study, we focus on GPT-2 Small (Radford et al., 2019), which has two induction heads in layer 5 (heads 5.1 and 5.5) (Wang et al., 2023). To distinguish between these two heads, we qualitatively inspect the top ten SAE features attributed to both heads (as in Section 4.1) and look for patterns. Glancing at the top features attributed to head 5.1 shows "long induction" features, defined as features that activate on examples of induction with at least two repeated prefix matches (e.g. completing "... ABC ... AB" with C).

We now confirm this hypothesis with independent lines of evidence that don't require SAEs. We first generate synthetic induction datasets with random repeated tokens of varying prefix lengths. For each dataset, we compute the induction score, defined as the average attention to the token which induction would suggest comes next, for both heads. We confirm that while both induction scores rise as we increase prefix length, head 5.1 has a much more dramatic phase change as we transition to long prefixes (i.e. $\geq 2$) (Figure 3a).

We also find and intervene on real examples of long prefix induction from the training distribution, corrupting them to only be one prefix by replacing the 2nd left repeated token (i.e 'A' in ABC ... AB) with a different, random token. We find that this intervention effectively causes head 5.1 to stop doing induction, as its average induction score falls from 0.55 to 0.05. Head 5.5, meanwhile, maintains an average induction score of 0.43 (Figure 3b).

Intuitively, head 5.1 can act as a "tie-breaker" when there are multiple candidates for induction. Consider the sequence " ... BD ... ABC ... AB". While head 5.5 might attend to both "D" and "C", head 5.1 can be used to more confidently predict the token "C". We provide evidence for this in Appendix Q, where we find that 5.1 has significant causal effect in boosting the predictions of tokens in cases of long prefix (but not short prefix) induction.

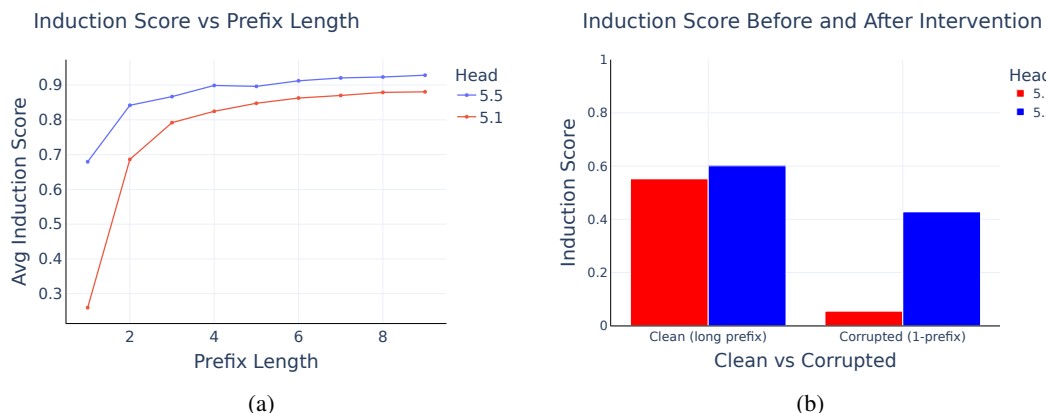

(a)                                                    (b)

Figure 3: Two lines of evidence that 5.1 specializes in long prefix induction, while 5.5 primarily does short prefix induction. In (a) we see that 5.1's induction score (Olsson et al., 2022) sharply increases from less than 0.3 to over 0.7 as we transition to long prefix lengths, while 5.5 already starts at 0.7 for short prefixes. In (b) we see that intervening on examples of long prefix induction from the training distribution causes 5.1 to essentially stop attending to that token, while 5.5 continues to show an induction attention pattern.

## 4.3 ANALYZING THE IOI CIRCUIT WITH ATTENTION OUTPUT SAEs

We now show that Attention Output SAEs are useful tools for circuit analysis. In the process, we also go beyond early work to find evidence that our SAEs find causally relevant intermediate variables. As a case study, we apply our SAEs to the widely studied Indirect Object Identification circuit (Wang et al., 2023), and find that our SAEs improve upon attention head interpretability based techniques from prior work.

The Indirect Object Identification (IOI) task (Wang et al., 2023) is to complete sentences like "After John and Mary went to the store, John gave a bottle of milk to" with " Mary" rather than " John". We refer to the repeated name (John) as S (the subject) and the non-repeated name (Mary) as IO (the indirect object). For each choice of the IO and S names, there are two prompt templates: one where the IO name comes first (the 'ABBA' template) and one where it comes second (the 'BABA' template).

Wang et al. (2023) analyzed this circuit by localizing and interpreting several classes of attention heads. They argue that the circuit implements the following algorithm:

1. Induction heads and Duplicate token heads identify that S is duplicated. They write information to indicate that this token is duplicated, as well as "positional signal" pointing to the S1 token.
2. S-inhibition heads route this information from S2 to END via V-composition (Elhage et al., 2021). They output both token and positional signals that cause the Name mover heads to attend less to S1 (and thus more to IO) via Q-composition (Elhage et al., 2021).
3. Name mover heads attend strongly to the IO position and copy, boosting the logits of the IO token that they attend to.

Although Wang et al. (2023) find that "positional signal" originating from the induction heads is a key aspect of this circuit, they don't figure out the specifics of what this signal is, and ultimately leave this mystery as one of the "most interesting future directions" of their work. Attention Output SAEs immediately reveal the positional signal by decomposing these activations into interpretable features. We find that rather than absolute or relative position between S tokens, the positional signal is actually whether the duplicate name comes after the " and" token that connects "John and Mary".

**Identifying the positional features:** To generate this hypothesis, we localized and interpreted causally relevant SAE features from the outputs of the attention layers that contain induction heads (Layers 5 and 6) with zero ablations. For now we focus on our Layer 5 SAE, and leave other layers

to Appendix N. In Appendix L we also evaluate that, for these layers, the SAE reconstructions are faithful on the IOI distribution, and thus viable for circuit analysis.

During each forward pass, we replace the L5 attention layer output activations with a sparse linear combination of SAE feature directions plus an error term, as in (1). We then zero ablate each feature, one at a time, and record the resulting change in logit difference between the IO and S tokens. This localizes three features that cause a notable decrease in average logit difference. See Appendix M for more details.

**Interpreting the "positional" features:** We then interpreted these causally relevant features. Shallow investigations of feature dashboards (see Section 3.1, Appendix A) suggests that all three of these fire on duplicate tokens, that were previously before or after " and" tokens (e.g. "I am a duplicate token that previously followed ' and'"). These feature interpretations motivated the hypothesis that the "positional signal" in IOI is solely determined by the position of the name relative to (i.e. before or after) the ' and' token.

**Confirming the hypothesis:** We now verify this hypothesis without reference to SAEs. We design a noising (defined in Heimersheim & Nanda (2024)) experiment that perturbs three properties of IOI prompts simultaneously, while preserving whether the duplicate name is before or after the ' and' token. Concretely, our counterfactual distribution makes the following changes:

1. *Replace* each name with another random name (removing "token signal" (Wang et al., 2023))
2. *Prepend* filler text (e.g. "It was a nice day") (corrupting absolute positions of all names)
3. *Add* filler text between S1 and S2 (corrupting the relative position between S tokens)

To demonstrate this procedure, first consider an example of a "clean" prompt: "When Alan and Alex got a drink at the store, Alex decided to give it to". After applying the changes above, we generate the "noised" prompt: "It was a nice day. When Ford and Joshua got a drink at the store, while the weather was nice, Joshua decided to give it to"

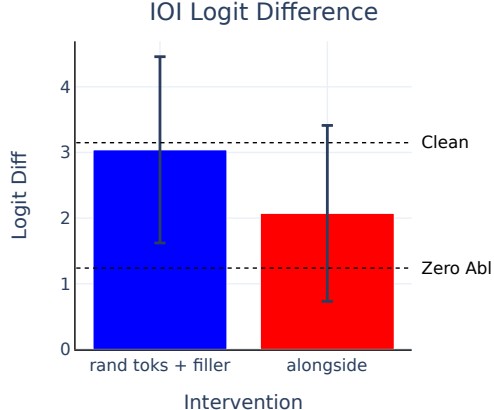

Despite being almost entirely different prompts, noising the attention layer outputs for both induction layers [5, 6] at the S2 position still recovers ~93% of average logit difference relative to zero ablating the outputs at this position (Figure 4).

One alternate hypothesis is that the positional signal is a more general emergent positional embedding (Nanda, 2023a) (e.g. "I am the second name in the sentence") that doesn't actually depend on the " and" token. We falsify this by noising attention outputs at layers [5,6] S2 position from a corrupted distribution which only changes " and" to the token " alongside". Note that this only corrupts one piece of information (the ' and') compared to the three corruptions above, yet we only recover ~43% of logit difference relative to zero ablation (Figure 4).

Figure 4: Results from two noising experiments on induction layers' attention outputs at S2 position.

## 5 RELATED WORK

**Mechanistic Interpretability.** Mechanistic interpretability research aims to reverse engineer neural network computations into human-understandable algorithms (Olah, 2022; Olah et al., 2020). Prior mechanistic interpretability work has identified computation subgraphs of models that implement tasks (Wang et al., 2023; Hanna et al., 2023; Lieberum et al., 2023), found interpretable, reoccurring model components over models of multiple sizes (Olsson et al., 2022; Gould et al., 2023), and reverse-engineered how toy tasks are carried out in small transformers (Nanda et al., 2023a; Chughtai et al., 2023). Some have successfully interpreted attention heads (McDougall et al., 2023;

Olsson et al., 2022; Wang et al., 2023), though the issue has been raised that heads are often polysemantic (Gould et al., 2023; Janiak et al., 2023), and may not be the correct unit of analysis (Jermyn et al., 2023). Our technique goes beyond prior work by decomposing the outputs of the entire attention layer into finer-grained linear features, without assuming that heads are the right unit of analysis.

Induction heads (Elhage et al., 2021) have been studied extensively by Olsson et al. (2022), who first observed that LLMs had many, seemingly redundant induction heads. Goldowsky-Dill et al. (2023) investigated two induction heads in a 2-layer attention-only model, and discovered the "long induction" (long-prefix induction) variant in *both* heads. In contrast, we find that two different induction heads *specialize* in long-prefix and short-prefix induction respectively in GPT-2 Small. In concurrent work, Ge et al. (2024) also find " and"-related feature in the IOI task. We causally verify the hypotheses of how " and" features behave in IOI and rule out alternative hypotheses.

**Classical Dictionary Learning.** Elad (2010) explores how both discrete and continuous representations can involve more representations than basis vectors, and surveys various techniques for extracting and reconstructing these representations. Traditional sparse coding algorithms (Olshausen & Field, 1997; Aharon et al., 2006) employ expectation-maximization, while contemporary approaches (Gregor & LeCun, 2010; Barello et al.) based on gradient descent and autoencoders have built upon these ideas.

**Sparse Autoencoders.** Motivated by the hypothesized phenomenon of supersition (Elhage et al., 2022), recent work has applied dictionary learning, specifically sparse autoencoders (Ng, 2011), to LMs in order to interpret their activations (Subramanian et al., 2017; Sharkey et al., 2022; Cunningham et al., 2023; Yun et al., 2023; Bricken et al., 2023; Templeton et al., 2024). Our feature interpretability methodology was inspired by Bricken et al. (2023), though we additionally study how features are computed upstream with direct feature attribution (Nanda et al., 2023b;c). Progress is rapid, with the following parallel work occurring within the last two months: Rajamanoharan et al. (2024) took a similar approach to our work and scaled attention output SAEs up to 7B models. Marks et al. (2024) also successfully used multiple types of SAEs including attention for finergrained circuit discovery with gradient based patching techniques. In contrast, we use both causal interventions and DFA, exploiting the linear structure of the attention mechanism.

## 6 CONCLUSION

In this work, we have introduced Attention Output SAEs, and demonstrated their effectiveness in decomposing attention layer outputs into sparse, interpretable features (Section 3). We have also highlighted the promise of Attention Output SAEs as a general purpose interpretability tool (Section 4). Our analysis identified novel and extant attention head motifs (Section 4.1), advanced our understanding of apparently 'redundant' induction heads (Section 4.2), and improved upon attention head circuit interpretability techniques from prior work (Section 4.3).

### 6.1 LIMITATIONS

Our work focuses on understanding attention outputs, which we consider to be a valuable contribution. However, we leave much of the transformer unexplained, such as the QK circuits (Elhage et al., 2021) by which attention patterns are computed. Further, though we scale up to a 2B model, our work was mostly performed on the 100M parameter GPT-2 Small model. Exploring Attention Output SAEs on larger models in depth is thus a natural direction of future work.

We also highlight some methodological limitations. While we try to validate our conclusions with multiple independent lines of evidence, our research often relies on qualitative investigations and subjective human judgment. Additionally, like all sparse autoencoder research, our work depends on both the assumptions made by the SAE architecture, and the quality of the trained SAEs. SAEs represent the sparse, linear components of models' computation, and hence may provide an incomplete picture of how to interpret attention layers (Rajamanoharan et al., 2024). Our SAEs achieve reasonable reconstruction accuracy (Table 1), though they are far from perfect.

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

## A    OPEN SOURCE SAE WEIGHTS AND FEATURE DASHBOARDS

Here we provide weights for all trained SAEs (Table 1) as well as the interface for feature dashboards that we used to evaluate feature interpretability discussed in Section 3.3. Weights are provided at an anonymous link: `https://huggingface.co/attention-saes-paper/attention-saes-paper-weights`

For GPT-2 Small, you can view feature dashboards for 30 randomly sampled feature per each layer here: `https://attention-saes-paper.github.io/attention-saes-paper-dashboards/attn-sae-gpt2-small-viz/`.

For our GELU-2L SAE trained on Layer 1 (the second layer), you can view dashboards for 50 randomly sampled features here: `https://attention-saes-paper.github.io/attention-saes-paper-dashboards/attn-sae-gelu-2l-viz/`.

To view the top 10 features attributed to all 144 attention heads in GPT-2 Small (as in Section 4.1) see here: `https://attention-saes-paper.github.io/attention-saes-paper-dashboards/gpt2-small-saes/`. We will release dashboards for the Gemma-2B SAE upon publication.

You can also view similar dashboards for any feature from all of our GPT-2 Small SAEs on neuronpedia (Lin & Bloom, 2023) here: `https://www.neuronpedia.org/gpt2-small/att-kk`.

Further, we introduce an interactive tool for exploring several attention SAEs throughout a model at `https://attention-saes-paper.github.io/attention-saes-paper-dashboards/circuit-explorer/` and discuss this more fully in Appendix R. Code will be released upon publication.

## B    SAE TRAINING: HYPERPARAMETERS AND OTHER DETAILS

Important details of SAE training include:

- **SAE Widths**. Our GELU-2L and Gemma-2B SAEs have width $16384$. All of our GPT-2 Small SAEs have width $24576$, with the exception of layers 5 and 7, which have width $49152$.

- **Loss Function**. We trained our Gemma-2B SAE with a different loss function than the SAEs from other models. For Gemma-2B we closely follow the approach from Olah et al. (2024), while for GELU-2L and GPT-2 Small, we closely follow the approach from Bricken et al. (2023).

- **Training Data**. We use activations from hundreds of millions to billions of activations from LM forward passes as input data to the SAE. Following Nanda (2023b), we use a shuffled buffer of these activations, so that optimization steps don't use data from highly correlated activations. For GELU-2L we use a mixture of 80% from the C4 Corpus (Raffel et al., 2023) and 20% code (`https://huggingface.co/datasets/NeelNanda/c4-code-tokenized-2b`). For GPT-2 Small we use OpenWebText (`https://huggingface.co/datasets/Skylion007/openwebtext`). For Gemma-2B we use `https://huggingface.co/datasets/HuggingFaceFW/fineweb`. The input activations have sequence length of 128 tokens for all training runs.

- **Resampling**. For our GELU-2L and GPT-2 Small SAEs we used *resampling*, a technique which at a high-level reinitializes features that activate extremely rarely on SAE inputs periodically throughout training. We mostly follow the approach described in the 'Neuron Resampling' appendix of Bricken et al. (2023), except we reapply learning rate warm-up after each resampling event, reducing learning rate to 0.1x the ordinary value, and, increasing it with a cosine schedule back to the ordinary value over the next 1000 training steps. Note we don't do this for Gemma-2B.

- **Optimizer hyperparameters**. For the GELU-2L and GPT-2 Small SAEs we use the Adam optimizer with $\beta_2 = 0.99$ and $\beta_1 = 0.9$ and a learning rate of roughly 0.001. For Gemma-

2B SAEs we also use the Adam optimizer with $\beta_2 = 0.999$ and $\beta_1 = 0.9$ and a learning rate of 0.00005.

## B.1 COMPUTE RESOURCES USED FOR TRAINING

Our GELU-2L SAE was trained on a single A6000 instance available from Vast AI[2] overnight. Our GPT-2 Small SAEs were each trained overnight on a single A100 instance also available from Vast AI. Our Gemma-2B SAE was also trained overnight on a single A100 instance from Paperspace[3].

The analyses described in the paper were performed on either an A6000 or A100 instance depending on memory bandwidth requirements. In no case were multiple machines or distributed tensors required for training or obtaining our experimental results. Most experiments take seconds or minutes, and all can be performed in under an hour.

The RDFA tool described in Appendix R is hosted on an A6000 instance available from `https://www.paperspace.com/deployments`.

## C  METHODOLOGY FOR FEATURE INTERPRETABILITY

To evaluate interpretability for Attention Output SAE features, we manually rate the interpretability of a set of randomly sampled SAE features. For each SAE, the two raters (paper authors) collectively inspected 30 randomly sampled features.[4].

To assess a feature, the rater determined if there was a clear explanation for the feature's behavior. The rater viewed the top 20 maximum activating dataset examples for that feature, approximate direct logit effects (i.e. $W_U W_O \mathbf{d}_i$), and randomly sampled activating examples from lower activation ranges (as in Bricken et al. (2023)). The dataset used for each SAE consists of 10 million tokens sampled from the SAE training distribution.

For each max activating dataset example, we also show the corresponding source tokens with the top direct feature attribution by source position (Section 2), and additionally show the weight based head attribution for all heads in that layer (Section 2). The raters used an interface based on an open source SAE visualizer library (McDougall, 2024) modified to support attention layer outputs (see Appendix A). Note that we filter out dead features (features that don't activate at least once in 100,000 inputs, sometimes also referred to as "ultra low frequency cluster") from our interpretability analysis. These features were excluded from the denominator in reporting percentage interpretable in Table 1.

The raters had a relatively high bar for labeling a feature as interpretable (e.g. noticing a clear pattern with all 20 max activating dataset examples, as well as throughout the randomly sampled activations). However, we note that this methodology heavily relies on subjective human judgement, and thus there is always room for error. We expect both false positives (e.g. the raters are overconfident in their interpretations, despite the feature actually being polysemantic) and false negatives (e.g. the raters might miss more abstract features that are hard to spot with our feature dashboards).

### C.1  CONFIDENCE INTERVALS FOR PERCENTAGE OF INTERPRETABLE FEATURES

In this section, we provide 95% confidence intervals for the percentage of features that are reported as interpretable in Table 1. For each layer, we treat the number of features that are interpretable as a binomial random variable with proportion of success $p$ (percentage interpretable) sampled over $n$ trials (number of features inspected).

The Clopper-Pearson interval $S_\leq \cap S_\geq$ provides an exact method for calculating binomial confidence intervals (Clopper & Pearson, 1934), with:

$$S_\leq := \left\{ p \middle| \mathbb{P}\left[\text{Bin}(n; p) \leq x\right] > \frac{\alpha}{2} \right\} \tag{5}$$

---

[2] `https://vast.ai/`

[3] `https://www.paperspace.com/`

[4] For GELU-2L we evaluated 50 randomly sampled features and didn't use DFA by source position

and

$$S_{\geq} := \left\{ p \middle| \mathbb{P}\left[\text{Bin}(n;p) \geq x\right] > \frac{\alpha}{2} \right\} \tag{6}$$

where $\alpha$ is the confidence level and $\text{Bin}(n;p)$ is the binomial distribution. Due to a relationship between the binomial distribution and the beta distribution, the Clopper–Pearson interval can be calculated (Thulin, 2014) as:

$$B\left(\frac{\alpha}{2}; x, n-x+1\right) < p < B\left(1 - \frac{\alpha}{2}; x+1, n-x\right) \tag{7}$$

where $x = np$ is the number of successes and $B(p; v, w)$ is the $p$th quantile of a beta distribution with shape $v$ and $w$. We present 95% confidence intervals ($\alpha = 0.025$) for Table 1 in Table 2.

Table 2: Confidence intervals for interpretability of Attention Output SAEs trained across multiple models and layers.

| Model | Layer | % Interp. | 95% CI |
|---|---|---|---|
| Gemma-2B (Gemma Team et al., 2024) | 6 | 66% | [47.2%, 82.7%] |
| GPT-2 Small | 0 | 97% | [82.2%, 99.9%] |
| GPT-2 Small | 1 | 87% | [69.3%, 96.2%] |
| GPT-2 Small | 2 | 97% | [82.8%, 99.9%] |
| GPT-2 Small | 3 | 77% | [57.7%, 90.1%] |
| GPT-2 Small | 4 | 97% | [82.8%, 99.9%] |
| GPT-2 Small | 5 | 80% | [61.4%, 92.3%] |
| GPT-2 Small | 6 | 77% | [57.7%, 90.1%] |
| GPT-2 Small | 7 | 70% | [50.6%, 85.3%] |
| GPT-2 Small | 8 | 60% | [40.6%, 77.3%] |
| GPT-2 Small | 9 | 77% | [57.7%, 90.1%] |
| GPT-2 Small | 10 | 80% | [61.4%, 92.3%] |
| GPT-2 Small | 11 | 63% | [43.9%, 80.1%] |
| GELU-2L (Nanda, 2022b) | 1 | 83% | [65.3%, 94.4%] |

## D  INDUCTION FEATURE DEEP DIVE: ANALYZING FALSE NEGATIVES

In this section we display in Figure 5 two random examples of false negatives identified during the sensitivity analysis from Section 3.3. To recap, these are examples where our proxy identified a case of board induction (i.e. "<token> board ... <token>"), but the board induction feature did not fire. We generally notice that while they technically satisfy the board induction pattern, "board" should clearly not be predicted as the next token. This is often because there are even stronger cases of induction for another token (fig. 5).

## E  INDUCTION FEATURE DEEP DIVE: EXPLAINING POLYSEMANTICITY AT
##    LOWER ACTIVATION RANGES

In Section 3.3 we noticed that while the upper parts of the activation spectrum clearly respond with high specificity to 'board' induction, there were also many false positives in the lower activation ranges (as in Bricken et al. (2023)), we believe these are expected for mundane reasons:

- *Imperfect proxy*: Manually staring at the false positives in the medium activation ranges reveals examples of fuzzy 'board' induction that weren't identified by our simple proxy.
- *Undersized dictionary*: Our GELU-2L SAE has a dictionary of roughly 16,000 features. We expect our model to have many more "true features" (note there are 50k tokens in the vocabulary). Thus unrecovered features may show up as linear combinations of many of our learned features.

```
<|BOS|> ("boardUpdate", self.__onBoardUpdate)
        self.connection.om.connect("onChallengeAdd", self.__onChallengeAdd)
        self.connection.om.connect("onOfferAdd", self.__onOfferAdd)
        self.connection.adm.connect("onAdjournmentsList", self.__onAdjournmentsList)
        self.connection.em.connect("onAmbiguousMove", self.__onAmbiguousMove)
        self.connection.em.connect("onIllegalMove", self.__onAmbiguousMove)
```

(a)

```
<|BOS|> exist"

@mock.patch("ubittool.cli.read_flash_hex", autospec=True)
def test_read_flash(mock_read_flash_hex, check_no_board_connected):
    """Test the read-flash command without a file option."""
    flash_hex_content = "Intel Hex lines here"
    mock_read_flash_hex.return_value = flash_hex_content
    runner = CliRunner()

    result = runner.invoke(cli.read_flash)

    assert "micro:bit flash hex will be output to
```

(b)

Figure 5: Two examples of false negatives for the board induction feature. The red highlight indicates that our proxy is active, but the board feature is not.

- *Superposition*: The superposition hypothesis (Elhage et al., 2022) suggests that models represent sparse features as non-orthogonal directions, causing interference. If true, we should expect some polysemanticity at the lower activation ranges by default.

We also agree with the following intuition from Bricken et al. (2023): "large feature activations have larger impacts on model predictions, so getting their interpretation right matters most". Thus we reproduced their expected value plots to demonstrate that most of the magnitude of activation provided by this feature comes from 'board' induction examples in Figure 2b.

| Positive logits | | Negative logits | | | Positive logits | | Negative logits | | | Positive logits | | Negative logits | |
|---|---|---|---|---|---|---|---|---|---|---|---|---|---|
| board | 2.26 | Lemmon | -1.29 | | ? | 3.26 | ˙˙ | -0.82 | | dogs | 3.25 | DRAM | -1.60 |
| boards | 1.92 | Rah | -1.21 | | )? | 3.02 | vart | -0.74 | | canine | 3.05 | IBM | -1.42 |
| Board | 1.80 | Nag | -1.17 | | ?" | 2.99 | assertEqual | -0.72 | | Dogs | 3.02 | disk | -1.32 |
| BO | 1.71 | Allah | -1.17 | | ?' | 2.98 | assertEquals | -0.70 | | puppy | 3.00 | UM | -1.29 |
| boarding | 1.68 | adul | -1.15 | | ?! | 2.93 | 書 | -0.68 | | dog | 2.96 | Disk | -1.29 |
| pin | 1.65 | Sul | -1.15 | | ?¡ | 2.75 | ;;;; | -0.67 | | Dog | 2.77 | quas | -1.28 |
| wire | 1.55 | × | -1.13 | | ?. | 2.68 | FromString | -0.66 | | pets | 2.76 | Nokia | -1.27 |
| oard | 1.55 | Ariz | -1.13 | | ?! | 2.67 | DISCLAIM | -0.65 | | veter | 2.72 | IBM | -1.26 |
| bo | 1.51 | PSA | -1.12 | | !? | 2.66 | unal | -0.64 | | Dog | 2.63 | Oracle | -1.25 |
| aptop | 1.37 | serotonin | -1.12 | | ?' | 2.65 | messages | -0.64 | | pupp | 2.50 | semiconduct | -1.25 |
| board | 1.36 | sulph | -1.11 | | ?), | 2.57 | AFP | -0.63 | | pup | 2.46 | MOS | -1.22 |
| chess | 1.36 | Garc | -1.11 | | ?", | 2.54 | belle | -0.63 | | breeds | 2.44 | semiconductor | -1.21 |
| accord | 1.35 | ApJ | -1.11 | | ?? | 2.52 | SON | -0.63 | | cats | 2.42 | transistor | -1.20 |
| clock | 1.33 | ös | -1.11 | | "? | 2.52 | mpt | -0.63 | | breed | 2.41 | Tun | -1.19 |
| pad | 1.32 | Mex | -1.10 | | "? | 2.48 | suspicions | -0.62 | | Veter | 2.41 | IMF | -1.19 |
| bian | 1.30 | urea | -1.09 | | ??? | 2.46 | Assert | -0.62 | | veterinary | 2.40 | partitioning | -1.18 |
| elo | 1.29 | weeds | -1.08 | | .? | 2.38 | USY | -0.61 | | pet | 2.36 | phonon | -1.18 |
| cell | 1.28 | Persian | -1.07 | | ?, | 2.37 | stdout | -0.61 | | Animal | 2.35 | AMD | -1.18 |
| bels | 1.28 | Zar | -1.06 | | ? | 2.33 | Meier | -0.61 | | animals | 2.29 | anisotropic | -1.17 |
| hips | 1.27 | marry | -1.05 | | ?) | 2.27 | writ | -0.61 | | paw | 2.28 | DDR | -1.16 |

(a)         (b)         (c)

Figure 6: Direct logit effects of individual features: We show the top and bottom 20 affected output tokens from "'board' is next by induction" (a) "in a question starting with 'Which'" (b) and "in text about pets" (c) features

## F  INDUCTION FEATURE DEEP DIVE: UPSTREAM COMPUTATION AND DOWNSTREAM EFFECTS

In Section 3.3 we found a monosemantic SAE feature that represents that the "board" token is next by induction. In this section we show that we can also understand its causal downstream effects, as well as how it's computed by upstream components.

We first demonstrate that the presence of this feature has an interpretable causal effect on the outputs: we find that this feature is primarily used to directly predict the "board" token. We start by analyzing the approximate direct logit effect: $W_U W_O \mathbf{d}_i$ where $\mathbf{d}_i$ is this feature direction. We clearly see that the "board" token is the top logit in Figure 6a.

This interpretation is also corroborated by feature ablation experiments. Across all activating dataset examples over 10 million tokens, we splice in our Attention Output SAE at Layer 1 of the model (the last layer of GELU-2L), ablate the board induction feature, and record the effect on loss. We find that 82% of total loss increase from ablating this feature is explained by examples where board is the correct next token.

Finally, we demonstrate that we can understand how this feature is computed by upstream components. We first show that this feature is almost entirely produced by attention head 1.6, an induction head (Nanda, 2022a). Over 10 million tokens, we compute the direct feature attribution by head (see (3)) for this feature. We find that head 1.6 stands out with 94% fraction of variance explained.

Going further upstream, we now show that 1.6 is copying prior "board" tokens to activate this feature. We apply DFA by source position (see Section 2) for all feature activations over 10 million tokens and record aggregate scores for each source token. We find that the majority of variance is explained by "board" source tokens. This effect is stronger if we filter for feature activations above a certain threshold, reaching over 99.9% at a threshold of 5, mirroring results from Bricken et al. (2023) that there's more polysemanticity in lower ranges. We note that this "copying" is consistent with our understanding of the induction (Olsson et al., 2022) algorithm.

## G  LOCAL CONTEXT FEATURE DEEP DIVE: IN QUESTION STARTING WITH "WHICH"

We now consider an "In questions starting with 'Which'" feature. We categorized this as one of many "local context" features: a feature that is active in some context, but often only for a short time, and which has some clear ending marker (e.g. a question mark, closing parentheses, etc).

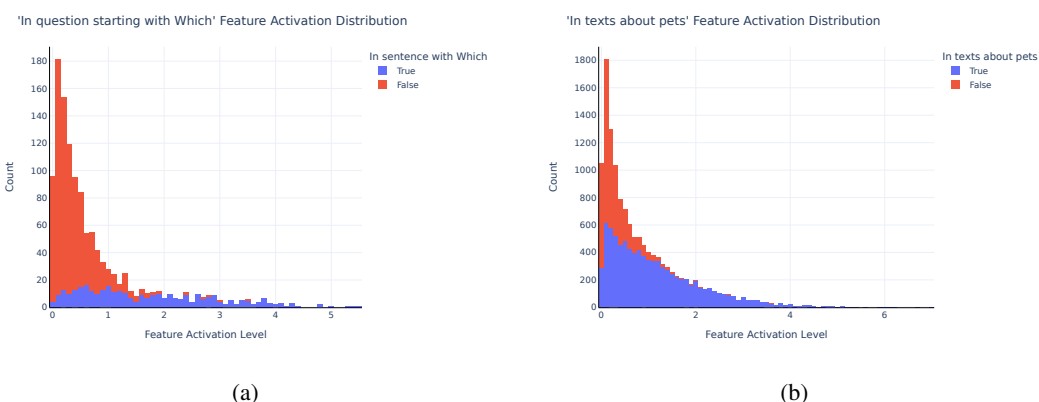

Figure 7: Specificity plots for "in question starting with 'Which'" (a) and "In text about pets" (b) features

Unlike the induction feature (Section 3.3), we also find that it's computed by multiple attention heads. The fact that our Attention SAEs extracted a feature relying on multiple heads, and made progress towards understanding it, suggests that we may be able to use Attention Output SAEs as a tool to tackle the hypothesized phenomenon of attention head superposition (Jermyn et al., 2023).

We first show that our interpretation is faithful over the entire distribution. We define a crude proxy that checks for the first 10 tokens after "Which" tokens, stopping early at punctuation. Similar to the induction feature, we find that this feature activates with high specificity to this context in the upper activation ranges, although there is polysemanticity for lower activations (Figure 7a).

We now show that the feature is computed by multiple heads in layer 1. Over 10 million tokens, we compute the direct feature attribution by head (3) for this feature. We find that head 3 heads have non-trivial (>10%) fraction of variance explained (Figure 8).

Despite this, we still get traction on understanding this feature, motivating attention SAEs as a valuable tool to deal with attention head superposition. We first understand the causal downstream effects of this feature. We find that it primarily "closes the question", by directly boosting the logits of question mark tokens (Figure 6b).

We also show that the heads in aggregate are moving information from prior "Which" tokens to compute this feature. We apply DFA by source position (aggregated across all heads) (see Section 2) for all feature activations over 10 million tokens and record aggregate scores for each source token. We find that "Which" source tokens explain >50% the variance, and over 95% of the variance if we filter for feature activations greater than 2, suggesting that the heads are moving this "Which" to compute the feature.

## H  HIGH LEVEL CONTEXT FEATURE DEEP DIVE: IN TEXT RELATED TO PETS

We now consider an "in a text related to pets" feature. This is one example from a family of 'high level context features' extracted by our SAE. High level context features often activate for almost the entire context, and don't have a clear ending marker (like a question mark). To us they appear qualitatively different from the local context features, like "in a question starting with 'Which'", which just activate for e.g. all tokens in a sentence.

We first show our interpretation of this feature is faithful. We define a proxy that checks for all tokens that occur after any token from a handcrafted set of pet related tokens ('dog', ' pet', ' canine', etc), and compare the activations of our feature to the proxy. Though the proxy is crude, we find that this feature activates with high specificity in this context in Figure 7b.

We show that we can understand the downstream effects of this feature. The feature directly boosts logits of pet related tokens ('dog', ' pet', ' canine', etc) in Figure 6c.

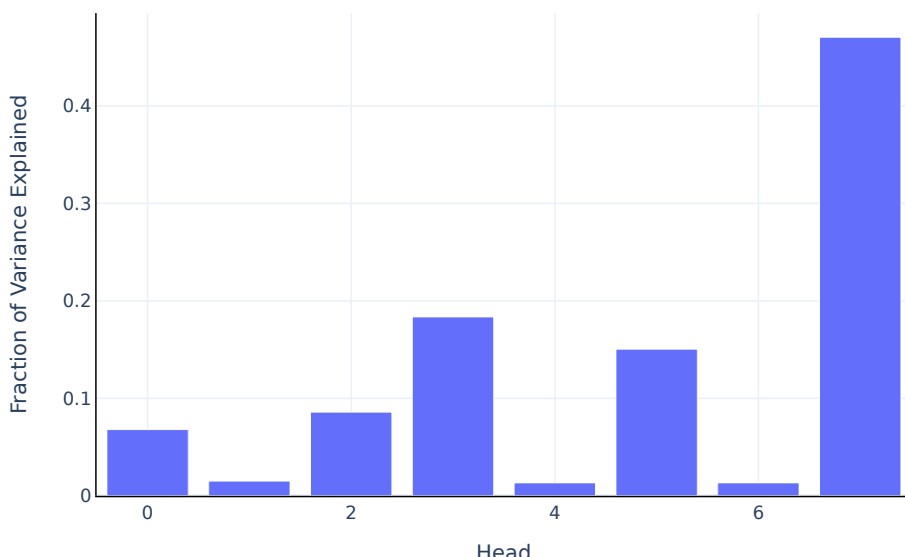

Figure 8: Fraction of variance of DFA by head explained for the "In a question starting with 'Which'" feature over 10 million tokens. We notice that this feature is distributed across multiple heads

In this case study we highlight that we were able to use techniques like direct feature attribution to learn that high level context features are natural to implement with a single attention head: the head can just look back for past "pet related tokens" ('dog', ' pet', ' canine', ' veterinary', etc), and move these to compute the feature.

We find that the top attention head is using the pet source tokens to compute the feature. We track the direct feature contributions from source tokens in a handcrafted set of pet related tokens ('dog', 'pet', etc) and compute the fraction of variance explained from these source tokens. We confirm that "pet" source tokens explain the majority of the variance, especially when filtering by higher activations, with over 90% fraction of variance explained for activations greater than 2.

# I   FURTHER DISCUSSION ON SAE FIDELITY EVALUATIONS

In Section 3 we claimed that our Attention Output SAEs are sparse, faithful, and interpretable and we provide evaluations of each SAE in Table 1 to support this claim. In this section we further discuss nuances of the fidelity evaluation, and how our SAEs compare to trained SAEs from other work.

We note that we evaluated fidelity with the cross entropy loss relative to zero ablation (4), which has a few potential pitfalls. First, some would argue that zero ablation may be too harsh a baseline, and that alternative baselines using mean ablation or resample ablation may be more principled. We choose to use zero ablation to stay consistent with prior work from Bricken et al. (2023), which made our preliminary results easier to evaluate.

Second, the zero ablation baseline makes it's hard to compare the quality of SAEs between other sites. Intuitively, zero ablating the residual stream should degrade performance much more than ablating a single attention layer or MLP, so we expect that SAEs trained on the residual stream will have much higher % CE recovered metrics, even if splicing in the residual stream SAE causes a

Table 3: Evaluations of sparsity, fidelity, and interpretability for Attention Output SAEs trained across multiple models and layers. Percentage of interpretable features were based on 30 randomly sampled live features inspected per layer.

| Model | Layer | L0 | % CE Rec. | % Int. | Clean CE Loss | SAE CE Loss | CE Delta | 0 Abl CE Loss |
|---|---|---|---|---|---|---|---|---|
| Gemma-2B | 6 | 92 | 75% | 66% | 2.6530 | 2.6810 | 0.0280 | 2.7670 |
| GPT-2 Small | 0 | 3 | 99% | 97% | 3.5803 | 3.6070 | 0.0266 | 7.2387 |
| GPT-2 Small | 1 | 20 | 79% | 87% | 3.5563 | 3.5595 | 0.0032 | 3.5722 |
| GPT-2 Small | 2 | 16 | 90% | 97% | 3.6049 | 3.6104 | 0.0055 | 3.6623 |
| GPT-2 Small | 3 | 15 | 84% | 77% | 3.6004 | 3.6062 | 0.0058 | 3.6365 |
| GPT-2 Small | 4 | 15 | 88% | 97% | 3.5565 | 3.5632 | 0.0067 | 3.6132 |
| GPT-2 Small | 5 | 20 | 85% | 80% | 3.5879 | 3.5968 | 0.0088 | 3.6469 |
| GPT-2 Small | 6 | 20 | 83% | 77% | 3.5930 | 3.6043 | 0.0113 | 3.6599 |
| GPT-2 Small | 7 | 19 | 84% | 70% | 3.5837 | 3.5942 | 0.0104 | 3.6530 |
| GPT-2 Small | 8 | 20 | 73% | 60% | 3.5896 | 3.6020 | 0.0124 | 3.6359 |
| GPT-2 Small | 9 | 21 | 82% | 77% | 3.6009 | 3.6117 | 0.0107 | 3.6617 |
| GPT-2 Small | 10 | 16 | 85% | 80% | 3.5658 | 3.5757 | 0.0099 | 3.6332 |
| GPT-2 Small | 11 | 8 | 88% | 63% | 3.5910 | 3.6047 | 0.0136 | 3.7042 |
| GPT-2 Small | All | | | 80% | | | | |
| GELU-2L | 1 | 11 | 88% | 83% | 2.7480 | 3.2271 | 0.4791 | 6.8098 |

much bigger jump in cross entropy loss. See Rajamanoharan et al. (2024) for thorough evaluations of trained SAEs across multiple sites. For this reason, we recommend practitioners additionally record the raw cross entropy loss numbers with and without the SAE spliced in, as in Table 3.

We also note that there is a trade off between sparsity and fidelity, and due to limited compute, we are likely far from pareto optimal. Recent work (Bloom, 2024; Templeton et al., 2024) has had success interpreting SAEs with higher numbers of features firing, although it's not clear what L0 we should target. For example, we might expect more features in the residual stream compared to an attention head, and we might expect larger models to compute more features than smaller models.

With this in mind, it's hard to compare our SAEs to across work that uses on different models and activation sites. When we trained our SAEs, we closely followed Bricken et al. (2023) as a reference. The MLP SAE from their work had a % CE recovered of 79%. They claimed that they generally targeted an L0 norm that is less than 10 or 20. Our SAEs have similar metrics, where we generally targeted and L0 of 20 with 80% CE loss recovered,

## J AUTOMATIC INDUCTION FEATURE DETECTION

In this section we automatically detect and quantify a large "<token> is next by induction" feature family from our GELU-2L SAE trained on layer 1. This represents  5% of the non-dead features in the SAE. This is notable, as if there are many "one feature per vocab token" families like this, we may need extremely wide SAEs for larger models.

Based on the findings of the "'board' is next by induction" feature (see Section 3.3), we surmised that there might exist more features with this property for different suffixes. Guided by this motivation, we were able to find 586 additional features that exhibited induction-like properties from our GELU-2L SAE. We intend this as a crude proof of concept for automated SAE feature family detection, and to show that there are many induction-like features. We think our method could be made significantly more rigorous with more time, and that it likely has both many false positives and false negatives.

While investigating the "board" feature, we confirmed that attention head 1.6 was an induction head. For each feature dashboard, we also generated a decoder weights distribution that gave an approximation of how much each head is attributed to a given feature. We then chose the following heuristic to identify additional features that exhibited induction-like properties:

**Induction Selection Heuristic.** For each feature, we compute the weight based head attribution score (2) to head 1.6. We consider features that have a head attribution score of at least 0.6 as induction feature candidates.

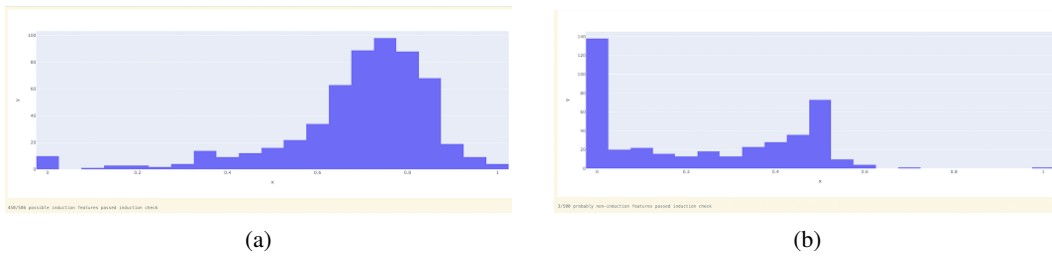

(a)                                                         (b)

Figure 9: *Automated Induction*: The features identified by our induction selection heuristic (a) selects 450/586 features that satisfy the induction behavior heuristic, whereas (b) the control group only selects 3.

Intuitively, given the normalized norms sum to 1, we expect features satisfying this property to primarily be responsible for producing induction behavior for specific sets of suffix tokens. In our case, we found 586 features that pass the above induction heuristic and are probably related to induction. We note that this is a conservative heuristic, as head 1.4 gets a partial score on the random tokens induction metric, and other heads may also play an induction-like role on some tokens, yet fail the random tokens test (Olsson et al., 2022).

We verified that these are indeed behaviorally related to induction using the following behavioral heuristic:

**Induction Behavior Heuristic**. For each feature, consider the token corresponding to the max positive boosted logit through the direct readout from $W_U W_O \mathbf{d}_i$. For a random sample of 200 examples that contain that token, identify which proportion satisfy:

1. For any given instance of the token corresponding to the max positive boosted logit for that feature, the feature does not fire on the first prefix of that token (i.e., the first instance of an "AB" pattern).
2. For any subsequent instances of the token corresponding to the max positive boosted logit for that feature occurring in the example, the feature activates on the preceding token (i.e. subsequent instances of an "AB" pattern).

We call the proportion of times the feature activates when it is expected to activate (on instances of A following the first instance of an AB pattern) the induction pass rate for the feature. The heuristic passes if the induction pass rate is $> 60\%$.

With the "board" feature, we saw that the token with the top positive logit boost passed this induction behavior heuristic: for almost every example and each bigram that ends with "board", the first such bigram did not activate the feature but all subsequent repeated instances did.

We ran this heuristic on the 586 features identified by the Induction Selection Heuristic against 500 features that have attribution $< 10\%$ to head 1.6 as a control group (i.e., features we would not expect to display induction-like properties as they are not attributed to the induction head). We found the Induction Behavior Heuristic to perform well at separating the features, as 450/586 features satisfied the $> 60\%$ induction pass rate. Conversely, only 3/500 features in the control group satisfied the $> 60\%$ induction pass rate.

## K    INVESTIGATING ATTENTION HEAD POLYSEMANTICITY

While the technique from Section 4.1 is not sufficient to prove that a head is monosemantic, we believe that having multiple unrelated features attributed to a head is evidence that the head is doing multiple tasks (i.e. exhibit polysemanticity (Elhage et al., 2022)). We also note that there is a possibility we missed some monosemantic heads due to missing patterns at certain levels of abstraction (e.g. some patterns might not be evident from a small sample of SAE features, and in other instances an SAE might have mistakenly learned some red herring features).

During our investigations of each head, we found 14 monosemantic candidates (i.e. all of the top 10 attributed features for these heads were closely related). This suggests that over 90% of the attention

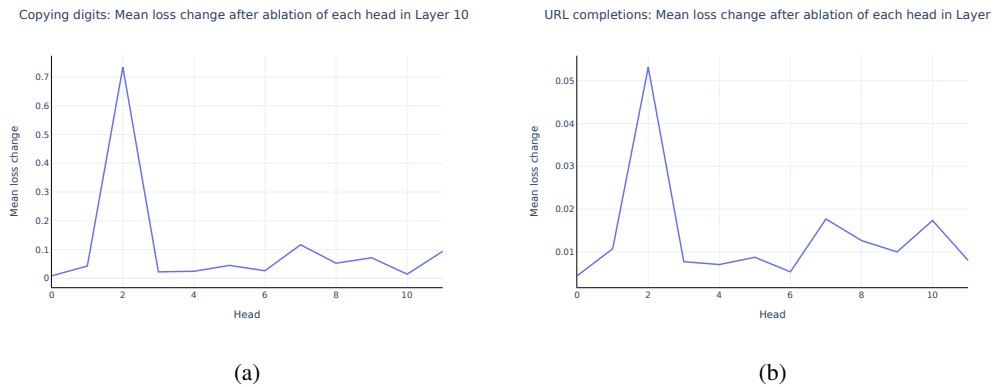

(a)                                   (b)

Figure 10: An indication of polysemanticity for head 10.2: both digits copying (a) and URL completion (b) behavior exhibits high mean loss change upon ablation relative to the other heads in layer 10.

heads in GPT-2 small are performing at least two different tasks. In Appendix K.1, we list notable heads that are plausibly monosemantic or have suggested roles based on this technique.

As one example of a validation of polysemanticity, Figure 10a and Figure 10b demonstrate two completely different behaviors[5] of 10.2 found in the top SAE features. Ablating this head and recording the mean change in loss on synthetic datasets for each task shows a clear jump for 10.2 relative to other heads, confirming that this head is involved in both tasks.

## K.1 POLYSEMANTIC ATTENTION HEADS IN GPT-2 SMALL

Based on the analysis in the previous section, we determined the statistics in Table 4 on polysemanticity within attention heads in GPT-2 Small.

Notably, the existence of any top features that do not belong to a conceptual grouping are sufficient evidence to dispute monosemanticity. On the other hand, all top features belonging to a conceptual grouping are weak evidence towards monosemanticity. Therefore, the results in this section form a lower bound on the percentage of attention heads in GPT-2 Small that are polysemantic.

Table 4: Proportion of heads exhibiting monosemantic versus polysemantic behavior.

| Head Type | Fraction of Heads |
| --- | --- |
| Plausibly monosemantic | 9.7% (14/144) |
| Plausibly monosemantic (minor exception) | 5.5% (8/144) |
| Plausibly bisemantic | 2.7% (4/144) |
| Polysemantic | 81.9% |

We say that a feature is *plausibly monosemantic* when all top 10 features were deemed conceptually related by our annotator, and *plausibly monosemantic (minor exception)* when all features were deemed conceptually related with only one or two exceptions. Finally, a feature is *plausibly bisemantic* when features were clearly in only two conceptual categories.

Finally, note that the line between polysemantic and monosemantic heads is a spectrum. For example, consider head 5.10: all top 10 SAE features look like context features, boosting the logits of tokens related to that context. However, our annotator conservatively labeled this head as polysemantic given that some of the contexts are unrelated. At a higher-level grouping, this head could plausibly be labeled a general monosemantic "context" head.

---

[5]By *digit copying* behavior, we refer to instances of boosting a specific digit found earlier in the prompt: for example, as in "Image 2/**8**... Image 5/**8**". By *URL completion*, we refer to instances of boosting plausible portions of a URL, such as the base64 tokens immediately following "pic.twitter.com/".

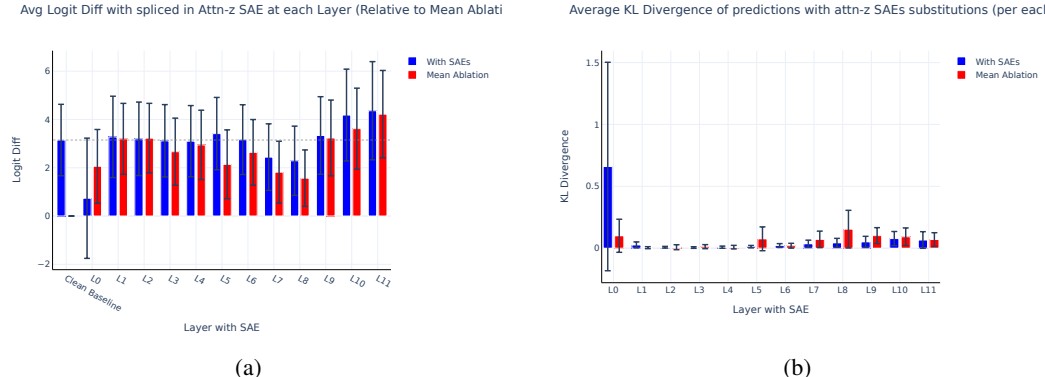

Figure 11: Evaluating each GPT-2 Small attention SAE on the IOI task. We splice in an Attention Output SAE for each layer and compare the resulting average logit difference (a) and KL divergence (b) to the model without SAEs. We also compare to a baseline where we mean ablate that layer's attention output from the ABC distribution (Wang et al., 2023). We generally observe that our SAEs from layers [1, 6] are sufficient, while our SAEs from layers [7,11] and 0 have noticeable reconstruction error.

## L    EVALUATING ALL GPT-2 SMALL SAES ON IOI

In this section we evaluate all of our GPT-2 Small attention SAEs on the IOI task. For each layer, we replace attention output activations with their SAE reconstructed activations and observe the effect on the average logit difference (Wang et al., 2023) between the correct and incorrect name tokens (as in Makelov et al. (2024)). We also measure the KL divergence between the logits of the original model and the logits of the model with the SAE spliced in. We compare the effect of splicing in the SAEs to mean ablating these attention layer outputs from the ABC distribution (as described in Wang et al. (2023), this is the IOI distribution but with three different names, rather than one IO and two subjects) to also get a rough sense of how necessary these activations are for the circuit.

We find that splicing in our SAEs at each of the early-middle layers [1, 6] maintains an average logit difference roughly equal to the clean baseline, suggesting that these SAEs are sufficient for circuit analysis. On the other hand, we see layers {0, 7, 8} cause a notable drop in logit difference. The later layers actually cause an increase in logit difference, but we think that these are likely breaking things based on the relatively high average KL divergence, illustrating the importance of using multiple metrics that capture different things. We suspect that these late layer SAEs might be missing features corresponding to the Negative Name Mover (Copy Suppression (McDougall et al., 2023)) heads in the IOI circuit, although we don't investigate this further.

Wang et al. (2023) identify many classes of attention heads spread across multiple layers. To investigate if our SAEs are systematically failing to capture features corresponding to certain heads, we splice in our SAEs for each of these cross-sections (similar to Makelov et al. (2024)).

For each role classified by Wang et al. (2023), we identify the set of attention layers containing all of these heads. We then replace the attention output activations for all of these layers with their reconstructed activations. Note that we recompute the reconstructed activations sequentially rather than patching all of them in at once. We do this for the following groups of heads:

- Duplicate Token Heads {0, 3}
- Previous Token Heads {2, 4}
- Induction Heads {5, 6}
- S-inhibition Heads {7, 8}
- (Negative) Name Mover Heads {9, 10, 11}

We again see promising signs that the early-middle layer SAEs (corresponding to the Induction and Previous Token Heads) seem sufficient for analysis at the feature level (Figure 12). Unfortunately, it's also clear that our SAEs are likely not sufficient to analyze the outputs of Layer 0 and the later

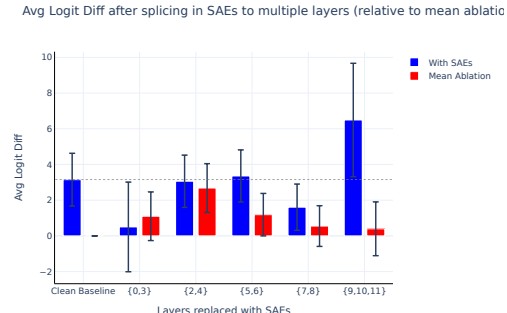 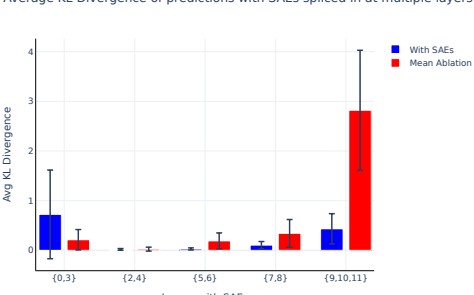

Figure 12: Evaluating cross sections of GPT-2 Small attention SAE on IOI. Here we splice in Attention Output SAEs for subsets of multiple layers in the same forward pass. Mirroring results from Appendix L, we find that the middle layers (corresponding the Previous Token and Induction Heads) are sufficient while later layers and Layer 0 have significant reconstruction error.

layers (S-inhibition Heads and (Negative) Name Mover Heads). Thus we are unable to study a full end-to-end feature circuit for IOI.

Why is there such a big difference between cross-sections? It is not clear from our analysis, but one hypothesis is that the middle layers contain more general features such as "I am a duplicate token", whereas the late layers contain niche name-specific features such as "The name X is next". Not only do we expect a much greater number of per-name features, but we also expect these features to be relatively rare, and thus harder for the SAEs to learn during training. We are hopeful that this will be improved by ongoing work on the science and scaling of SAEs (Nanda et al., 2024; Rajamanoharan et al., 2024; Olah et al., 2024).

# M   IOI CIRCUIT ANALYSIS: LAYER 5 "POSITIONAL" FEATURES

In this section, we describe how we identified and interpreted the causally relevant "positional" features form L5 (Section 4.3).

As mentioned, we first identify these features by zero ablating each feature one at a time and recording the resulting change in logit difference. Despite there being hundreds of features that fire at this position at least once, zero ablations narrow down three features that cause an average decrease in logit diff greater than 0.2. Note that ablating the error term has a minor effect relative to these features, corroborating our evaluations that our L5 SAE is sufficient for circuit analysis (Appendix L). We distinguish between ABBA and BABA prompts, as we find that the model uses different features based on the template (Figure 13a). We also localize the same three features when path patching features out of the S-inhibition head's (Wang et al., 2023) values, suggesting that these features are meaningfully V-composing (Elhage et al., 2021) with these heads, as the analysis from Wang et al. (2023) would suggest (Figure 13b). We find that features L5.F7515 and L5.F27535 are the most important for the BABA prompts, while feature L5.F44256 stands out for ABBA prompts.

We then interpreted these causally relevant features. Shallow investigations of feature dashboards (see Section 3.1, Appendix A) suggests that all three of these fire on duplicate tokens, and all have some dependence on prior " and" tokens. We hypothesize that the two BABA features are representing "I am a duplicate token that previously preceded ' and'" features, while the ABBA feature is "I am a duplicate token that previously followed ' and'". Note we additionally find similar causally relevant features from the induction head in Layer 6 and the duplicate token head in Layer 3 described in Appendix N. The features motivate the hypothesis that the "positional signal" in IOI is solely determined by the position of the name relative to (i.e. before or after) the ' and' token.

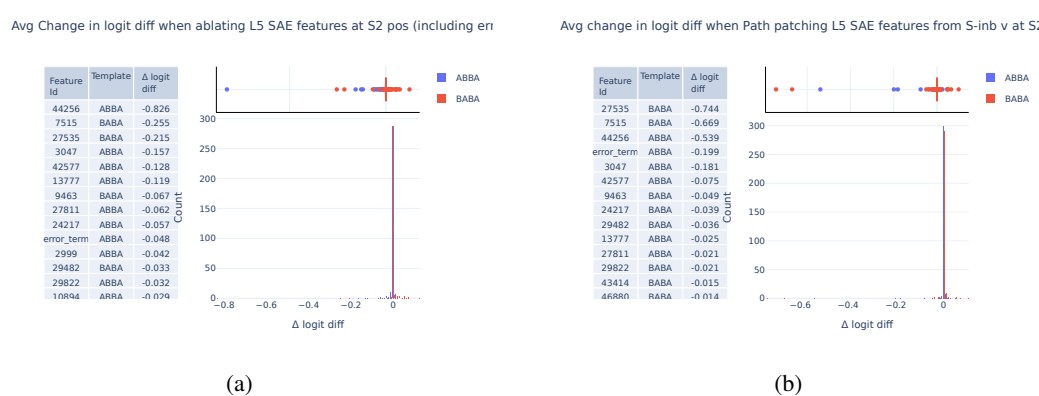

Figure 13: On the IOI (Wang et al., 2023) task, we identify causally relevant features from the layer 5 features with both zero ablations (a) and path patching (b) from the S-inhibition head values.

## N   IOI CIRCUIT ANALYSIS: FINDING AND INTERPRETING CAUSALLY RELEVANT FEATURES IN OTHER LAYERS

In addition to the L5 attention SAE features we showcase in Section 4.3, we also find features in other layers that seem to activate on duplicate tokens depending on their relative position to an " and" token. Note we didn't seek out features with these properties: these were all identified as the top causally relevant features via zero ablations for their respective layers (at the S2 position).

In Layer 3, a layer with duplicate token head 3.0 (Wang et al., 2023), we identify L3.F7803: "I am a duplicate token that was previously followed by 'and'/'or'" (Figure 14).

In L ayer 6, a layer with induction head 6.9 (Wang et al., 2023), we find two subltly different features:

- L6.F17410: "I am a (fuzzy) duplicate token that previously preceded ' and'".
- L6.F13836: "I am a duplicate name that previously preceded ' and'."

All of these features can be viewed with neuronpedia (Lin & Bloom, 2023): `https://www.neuronpedia.org/gpt2-small/att-kk`.

## O   IOI CIRCUIT ANALYSIS: APPLYING SAES TO QK CIRCUITS: S-INHIBITION HEADS SOMETIMES DO IO-BOOSTING

In addition to answering an open question about the positional signal in IOI (Wang et al., 2023) (Section 4.3), we also can use our SAEs to gain deeper insight into how these positional features are used downstream. Recall that Wang et al. (2023) found that the induction head outputs V-compose (Elhage et al., 2021) with the S-inhibition heads, which then Q-compose (Elhage et al., 2021) with the Name Mover heads, causing them to attend to the correct name. Our SAEs allow us to zoom in on this sub-circuit in finer detail.

We use the classic path expansion trick from Elhage et al. (2021) to zoom in on a Name Mover head's QK sub-circuit for this path:

$$\mathbf{x}_{\text{attn}} W_{\text{OV}}^{S-\text{inb}} W_{\text{QK}}^{NM} (\mathbf{x}_{\text{resid}})^T$$

Where $\mathbf{x}_{\text{attn}}$ is the attention output for a layer with induction heads, $W_{\text{OV}}^{S-\text{inb}}$ is the OV matrix (Elhage et al., 2021) for an S-inhibition head, $W_{\text{QK}}^{NM}$ is the QK matrix (Elhage et al., 2021) for a name mover head, and $\mathbf{x}_{\text{resid}}$ is the residual stream which is the input into the name mover head. For this case study we zoom into induction layer 5, S-inhibition head 8.6, and name mover head 9.9 (Wang et al., 2023).

Figure 14: We show max activating dataset examples and the corresponding top DFA by source position for L3.F7803 in GPT-2 Small, a causally relevant feature in the IOI task. We interpret this feature as representing "I am a duplicate token that was previously followed by 'and'/'or'". Notice that it seems to fire on duplicated tokens, and the previous duplicate (highlighted in blue) is almost always preceded by 'and'/'or'.

While the $\mathbf{x}_{\text{attn}}$ and $\mathbf{x}_{\text{resid}}$ terms on each side are not inherently interpretable units (e.g. the residual stream is tracking a large number of concepts at the same time, cf the superposition hypothesis (Elhage et al., 2022)), SAEs allow us to rewrite these activations as a weighted sum of sparse, interpretable features plus an error term (see 1).

This allows us to substitute both the $\mathbf{x}_{\text{attn}}$ and $\mathbf{x}_{\text{resid}}$ (using residual stream SAEs from Bloom (2024)) terms with their SAE decomposition. We then multiply these matrices to obtain an interpretable lookup table between SAE features for this QK subcircuit: Given that this S-inhibition head moves some Layer 5 attn SAE feature to be used as a Name Mover query, how much does it "want" to attend to a residual stream feature on the key side?

We find that the attention scores for this path can be explained by just a handful of sparse, interpretable pairs of SAE features. We zoom into the attention score from the END destination position (i.e. where we evaluate the model's prediction) to the Name2 source position (e.g. ' Mary' in " When John and Mary . . . ").

We observe that these heatmaps are almost entirely explained by a handful of reoccurring SAE features. On the query side we see the same causally relevant Attention SAE features previously identified by ablations: L5.F7515 and L5.F27535 ("I am a duplicate that preceded ' and'") for BABA prompts while ABBA prompts show L5.F44256 and L5.F3047 ("I am a duplicate that followed ' and'"). On the key side we also find just 2 common residual stream features doing most of the heavy lifting: L9.F16927 and L9.F4444 which both appear to activate on names following " and".

We also observe a stark difference in the heatmaps between prompt templates: while these pairs of features cause a decrease in attention score on the ABBA prompts, we actually see an increase in

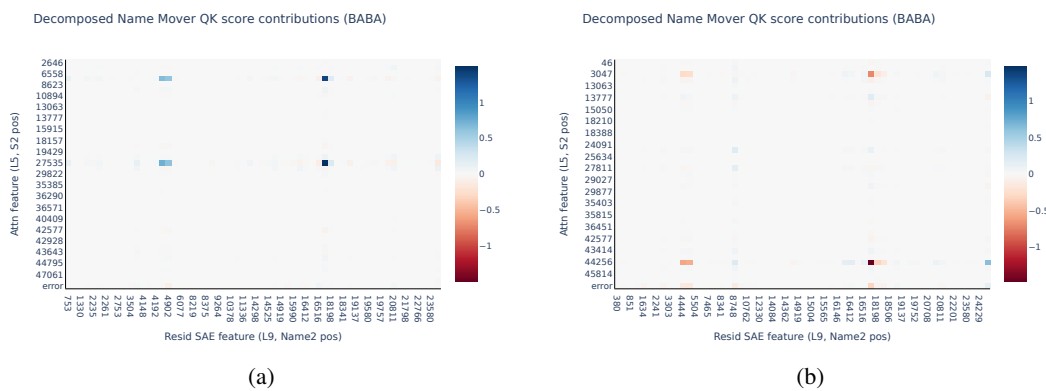

(a)                        (b)

Figure 15: We decompose the attention score from the END destination position for the Name2 source position into sparse, interpretable pairs of attention SAE features and residual stream SAE features. We notice that these features (a) boost the attention score to this position an BABA prompt, but (b) inhibit it on an ABBA prompt.

attention score on the BABA prompts. This suggests a slightly different algorithm between the two templates. On ABBA prompts, the S-inhibition heads move "I am a duplicate following 'and'" to "don't attend to the name following ' and'" (i.e. S-inhibition), while in BABA prompts it moves "I am a duplicate before ' and'" to "attend to the name following and". This suggests that the S-inhibition heads are partially doing "IO-boosting" on these BABA prompts.

To sanity check that our SAE based interpretations are capturing something real about this QK circuit, we compute how much of the variance in these heatmaps is explained by just these 8 pairs of interpretable SAE features. We find that these 8 pairs of SAE features explain 62% of the variance of the scores over all 100 prompts. For reference, all of the entries that include at least one error term (for both the attention output and residual stream SAEs) only explain approximately 15% of the variance.

## P  ADDITIONAL FEATURE FAMILIES IN GPT-2 SMALL

In this section we present new feature families that we found in GPT-2 Small, but did not find in the GELU-2L SAE[6]. This suggests that SAEs are a useful tool that can provide hints about fundamentally different capabilities as we apply them to bigger models.

**Duplicate Token Features.** In our Layer 3 SAE, we find many features which activate on repeated tokens. However, unlike induction features (Section 3.3), these have high direct feature attribution (by source position) to the previous instance of that token (rather than the token following the previous instance).

We also notice that the norms of the decoder weights corresponding to head 3.0, identified as a duplicate token head by Wang et al, stand out. This shows that, similar to the induction feature, we can use weight based attribution (2) to heads with previously known mechanisms to suggest the existence of certain feature families and vice versa.

**Successor Features.** In our Layer 9 SAE, we find features that activate in sequences of numbers, dates, letters, etc. The DFA by source position also suggests that the attention layer is looking back at the previous item(s) to compute this feature (Figure 16a).

The top logits of these features are also interpretable, suggesting that these features boost the next item in the sequence. Finally, the decoder weight norms also suggest that they heavily rely on head 9.1, a successor head in GPT-2 Small.

---

[6]Note we didn't exhaustively check every GELU-2L feature. However we never came across these in all of our analysis, whereas we quickly discovered these when looking at random features from GPT-2 Small

(a) L9.F18, a succession feature (Gould et al., 2023; Michaud et al., 2024)

(b) L10.F1610, a suppression feature (McDougall et al., 2023)

Figure 16: Two notable feature families extracted from the attention outputs of GPT-2 Small.

**Name Mover Features.** In the later layers, we also find features that seem to predict a name in the context. The defining characteristic of these features is a very high logit boost to the name. We also see very high DFA by source position to the past instances of this name in the context. Once again, our decoder weights also suggest that heads 9.9 and 9.6 are the top contributors of the feature, which were both identified as name mover heads by Wang et al. (2023).

We find a relatively large number of name movers within our shallow investigations of the first 30 random features, suggesting that this might explain a surprisingly large fraction of what the late attention layers are doing.

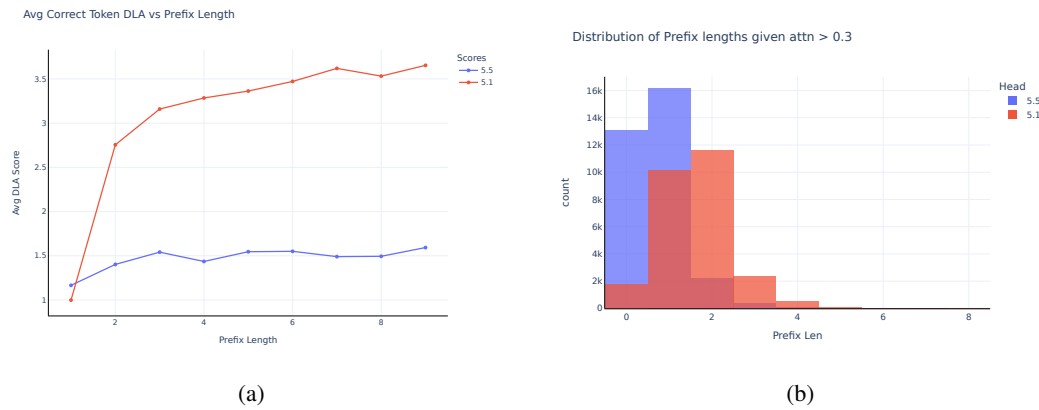

(a)                                        (b)

Figure 17: Two additional evidence that in GPT-2 Small, head 5.1 specializes in long prefix induction whereas head 5.5 does standard induction. (a) Head 5.1's direct logit attribution to the token that is next by induction increases sharply for long prefixes. (b) For examples where heads 5.1 and 5.5 are attending strongly to some token, head 5.1 is mostly performing long prefix induction whereas 5.5 is mostly performing short prefix induction.

**Suppression Features.**    Finally, in our layer 10 SAE we find suppression features (Figure 16b). These features show very low negative logits to a token in the context, suggesting that they actually seem to suppress these predictions. We use DFA to confirm that these features are being activated by previous instances of these tokens. Our decoder weights also identify head 10.7 as the top contributing head, the same head identified to do copy suppression by McDougall et al. (2023).

**N-gram Features.**    All of the features we have shown so far are related to previously studied behaviors, making them easier to spot and understand. We now show that we can also use our SAE to find new, surprising information about what attention layers have learned. We find a feature from Layer 9 that seems to be completing a common n-gram, predicting the "half" in phrases like "<number> and a half".

Though n-grams may seem like a simple capability, it's worth emphasizing why this is surprising. The intuitive way to implement n-grams would involve some kind of boolean AND (eg the current token is "and" AND the previous token is a number). Intuitively, this appears it would make sense to implement in MLPs and not in attention.

# Q    ADDITIONAL LONG PREFIX INDUCTION EXPERIMENTS

Here we provide two additional lines of evidence to show that in GPT-2 Small, 5.1 specializes in "long prefix induction", while 5.5 does "short prefix induction". Note we that we do not use SAEs in this section, but the original hypothesis was motivated by our SAEs (see Section 4.2).

We first check each head's average direct logit attribution (DLA) (Olsson et al., 2022) to the correct next token as a function of prefix length. We again see that head 5.1's DLA sharply increases as we enter the long prefix regime, while head 5.5's DLA remains relatively constant (Figure 17a).

We then confirmed that these results hold on a random sample of the training distribution. We first filter for examples where the heads are attending non-trivially to some token[7] (i.e. not just attending to BOS), and check how often these are examples of n-prefix induction. We find that head 5.1 will mostly attend to tokens in long prefix induction, while head 5.5 is mostly doing normal 1-prefix induction (Figure 17b).

---

[7]We show a threshold of 0.3. The results generally hold for a range of thresholds.

## R    RECURSIVE DIRECT FEATURE ATTRIBUTION

In this section, we expand on more manual circuit analysis techniques from Section 4.3. We can extend the DFA technique described in Section 2 by taking advantage of the fact that once we have frozen attention patterns and LayerNorms scales, there is a linear contribution from (1) different token position residual streams, (2) upstream model components, and (3) upstream Attention Output SAE decoder weight features to downstream Attention Output SAE features. This enables us to perform a fine-grained decomposition of attention SAEs recursively through earlier token position residual streams and upstream components. We call this technique Recursive DFA (RDFA).

We open-source a tool that enables performing this kind of recursive DFA on arbitrary prompts for GPT-2 Small. We currently only support this recursive attribution from attention to attention components, as we cannot pass upstream linearly through MLPs due to the element wise non-linear activation function. The tool is available at: `https://attention-saes-paper.github.io/attention-saes-paper-dashboards/circuit-explorer/`.

### R.1    RECURSIVE DIRECT FEATURE ATTRIBUTION (RDFA)

We use our Attention Output SAEs from Section 3.2 and residual stream SAEs from Bloom (2024) to repeatedly attribute SAE feature activation to upstream SAE feature outputs, all the way back to the input tokens for an arbitrary prompt. We provide the full Recursive DFA algorithm in Appendix R.

In Table 5, we provide a full description of recursive direct feature attribution (RDFA).

Table 5: Recursive direct feature attribution (RDFA)

| Step | Operation |
|------|-----------|
| 1. Choose an attention SAE feature index $i$: | $f_i^{\text{pre}}(\mathbf{z}_{\text{cat}}) = \mathbf{z}_{\text{cat}} \cdot W_{\text{enc}}[:, i]$ |
| 2. Compute DFA by source position: | $\mathbf{z}_{\text{dest}} = \sum_{\text{src}=0}^{\text{dest}} A[\text{dest}, \text{src}] \cdot \mathbf{v}_{\text{src}}$ |
| 3. Compute DFA by residual stream feature at source position $S$: | $\mathbf{v}_{\text{src}} = W_V \text{LN}_1(\mathbf{x}_{\text{resid}}) =$ $W_V \text{LN}_1 \left( \sum_{i=0}^{d_{\text{sae}}} f_i(\mathbf{x}_{\text{resid}})\mathbf{d}_i + \varepsilon(\mathbf{x}_{\text{resid}}) + \mathbf{b} \right)$ |
| 4. Compute DFA by upstream component for each resid feature: | $\mathbf{x}_{\text{resid}} = \mathbf{e} + \mathbf{p} + \sum_{i=0}^{L-1} \mathbf{x}_{\text{attn}} + \sum_{i=0}^{L-1} \mathbf{x}_{\text{mlp}}$ |
| 5. Decompose upstream attention layer outputs into SAE features: | $\mathbf{x}_{\text{attn}} = \sum_{i=0}^{d_{\text{sae}}} f_i(\mathbf{x}_{\text{attn}})\mathbf{d}_i + \varepsilon(\mathbf{x}_{\text{attn}}) + \mathbf{b}_{\text{dec}}$ |
| 6. Recurse: | Take one of the attention SAE features from the previous step and a prefix of our prompt at $S$. Then, treat $S$ as the destination position, and go back to step 1. |

## S    NOTABLE HEADS IN GPT-2 SMALL

As a continuation of Appendix S, we describe the results of manually inspecting the most salient features for all 144 attention heads to examine the role of every attention head in GPT-2 Small. As in Section 2, we apply equation 2 to identify the top ten features by decoder weight attribution to determine which features are most attributed to a given head. We then identify conceptual groupings that are exhibited in these features.

## S.1 Limitations on interpreting all heads in GPT-2 Small

We note that this methodology is a rough heuristic to get a sense of the most salient effects of a head and likely does not capture their role completely.

Through this technique we discover a wide range of previously unidentified behaviors. To validate that our technique captures legitimate phenomena rather than spurious behaviors, we verified that our interpretations are consistent with previously studied heads in GPT-2 Small. These include induction heads (Olsson et al., 2022; Kissane, 2023), previous token heads (Voita et al., 2019; Kissane, 2023), successor heads (Gould et al., 2023) and duplicate token heads (Wang et al., 2023; Kissane, 2023).

## S.2 Overview of attention heads in layers in GPT-2 Small

Broadly, we observe that top features attributed to heads become more abstract towards the middle layers of the model before tapering off to syntactic features in late layers:

- *Layers 0-3* exhibit primarily syntactic features (single-token features bigram features) and secondarily on specific verbs and entity fragments. Some context tracking features are also present.

- From *layer 4* onwards, features that activate on more complex grammatical structure are expressed, including families of related active verbs, prescriptive and active assertions, and some entity characterizations. Some single-token and bigram syntactic features continue to be present.

- In *layers 5-6*, we identify 2 out of the 3 known induction heads Goldowsky-Dill et al. (2023) in these layers based on our features. However, the rest of these layers is less interpretable through the lens of SAE features.

- In *layers 7-8*, increasingly more complex concept feature groups are present, such as phrasings related to specific actions taken, reasoning and justification related phrases, grammatical compound phrases, and time and distance relationships.

- *Layer 9* expressed some of the most complex concepts, with heads focused on specific concepts and related groups of concepts.

- *Layer 10* exhibited complex concept groups, with heads focused on assertions about a physical or spatial property, and counterfactual and timing/tense assertions.

- The last *layer 11* exhibited mostly grammatical adjustments, some bigram completions and one head focused on long-range context tracking.

Although the above summarizes what was distinctive about each layer, later layers continued to express syntactic features (e.g. single token features, URL completion) and simple context tracking features (e.g. news articles).

## S.3 Notable attention heads in GPT-2 Small

Appendix S.3 lists some notable attention heads across all layers of GPT-2 Small.

p0.1p0.4p0.45

| Layer | Feature groups / possible roles | Notable Heads |
|-------|--------------------------------|---------------|

**Table 5 – continued from previous page**

| Layer | Feature groups / possible roles | Notable Heads |
|-------|--------------------------------|---------------|

Continued on next page

0 Single-token ("of").
bigram features (following "S").
Micro-context features (cars, Apple tech, solar)  H0.1 Top 6 features are all variants capturing "of".
H0.5: Identified as duplicate token head from 9/10 features
H0.9: Long range context tracking family (headlines, sequential lists).

---

1 Single-token (Roman numerals)
bigram features (following "L")
Specific noun tracking (choice, refugee, gender, film/movie)  H1.5*: Succession (Gould et al., 2023; Michaud et al., 2024) or pairs related behavior
H1.8: Long range context tracking with very weak weight attribution

---

2 Short phrases ("never been...")
Entity Features (court, media, govt)
bigram & tri-gram features ("un-") Physical direction and logical relationships ("under") Entities followed by what happened (govt)  H2.0: Short phrases following a predicate (e.g., not/just/never/more)
H2.3: Short phrases following a quantifier (both, all, every, either), or spatial/temporal predicate (after, before, where)
H2.5: Subject tracking for physical directions (under, after, between, by), logical relationships (then X, both A and B)
H2.7: Groups of context tracking features
H2.9*: Entity followed by a description of what it did

---

3 Entity-related fragments (""world's X")
Tracking of a characteristic (ordinality or extremity)
Single-token and double-token (eg)
Tracking following commands (while, though, given)  H3.0: Identified as duplicate token head from 8/10 features
H3.2*: Subjects of predicates (so/of/such/how/from/as/that/to/be/by)
H3.6: Government entity related fragments, extremity related phrases
H3.11: Tracking of ordinality or entirety or extremity

---

4 Active verbs (do, share)
Specific characterizations (the same X, so Y)
Context tracking families (story highlights)
Single-token (predecessor)  H4.5: Characterizations of typicality or extremity
H4.7: Weak/non-standard duplicate token head
H4.11*: Identified as a previous token head based on all features

---

5 Induction (F)  H5.1: Long prefix Induction head
H5.5: Induction head

---

6 Induction (M)
Active verbs (want to, going to)
Local context tracking for certain concepts (vegetation)  H6.3:: Active verb tracking following a comma
H6.5: Short phrases related to agreement building
H6.7: Local context tracking for certain concepts (payment, vegetation, recruiting, death)
H6.9*: Induction head
H6.11: Suffix completions on specific verb and phrase forms

---

7 Induction (al-)
Active verbs (asked/needed)

Reasoning and justification phrases (because, for which)  H7.2*: Non-standard induction
H7.5: Highly polysemantic but still some groupings like family relationship tracking
H7.8: Phrases related to how things are going or specific action taken (decision to X, issue was Y, situation is Z)
H7.9: Reasoning and justification related phrasing (of which, to which, just because, for which, at least, we believe, in fact)
H7.10*: Induction head

---

8  Active verbs ("hold")
Compound phrases (either)
Time and distance relationships
Quantity or size comparisons or specifiers (larger/smaller)
URL completions (twitter)  H8.1*: Prepositions copying (with, for, on, to, in, at, by, of, as, from)
H8.5: Grammatical compound phrases (either A or B, neither C nor D, not only Z)
H8.8: Quantity or time comparisons/specifiers

---

9  Complex concept completions (time, eyes)
Specific entity concepts
Grammatical relationship joiners (between)
Assertions about characteristics (big/large)  H9.0*: Complex tracking on specific concepts (what is happening to time, where focus should be, actions done to eyes, etc.)
H9.2: Complex concept completions (death, diagnosis, LGBT discrimination, problem and issue, feminism, safety)
H9.9*: Copying, usually names, with some induction
H9.10: Grammatical relationship joiners (from X to, Y with, aided by, from/after, between)

---

10  Grammatical adjusters
Physical or spatial property assertions
Counterfactual and timing/tense assertions (would have, (hoped that))
Certain prepositional expressions (along, (under))
Capital letter completions ('B')  H10.1: Assertions about a physical or spatial property (up/back/down/over/full/hard/soft)
H10.4: Various separator characters for quantifiers (colon for time, hyphen for phone, period for counters)
H10.5: Counterfactual and timing/tense assertions (if/than/had/since/will/would/until/has X/have Y)
H10.6: Official titles
H10.10*: Capital letter completions with some context tracking (possibly non-standard induction)
H10.11: Certain conceptual relationships

---

11  Grammatical adjustments
bigrams
Capital letter completions
Long range context tracking  H11.3: Late layer long range context tracking, possibly for output confidence calibration

