# OpenReview forum: "Interpreting Attention Layer Outputs with Sparse Autoencoders"
_ICLR.cc/2025/Conference — Submitted to ICLR 2025_

### Official Review · Reviewer_kV6P · 2024-10-30

**Soundness:** 1
**Presentation:** 2
**Contribution:** 1
**Rating:** 3
**Confidence:** 4

**Summary:**

This paper applies sparse autoencoders, a popular tool in mechanistic interpretability, to interpret attention layer outputs as opposed to residual or MLP layers in large language models, as is usually done. The paper argues that this is a useful tool to understand transformer circuits, based on qualitative analysis. The paper applies its methods to understand induction heads, and study the Indirect Object Indentification (IOI) task.

**Strengths:**

+ The paper is easy to read, and adequately discusses the recent related work on mechanistic interpretability, particularly sparse autoencoders

+ The paper attempts to also independently verify the SAE-based insights obtained via non-SAE-based analysis, i.e., by using both correlational and intervention-based evidence

**Weaknesses:**

- A critical drawback of this paper is the non-systematic and ad-hoc nature of its analysis. For example, the paper includes phrases such as "we manually inspect these features for all 144 attention heads in GPT-2 Small. Broadly, we observe that features become more abstract in middle-layer heads and then taper off in abstraction at late layers", and "We use our weight-based head attribution technique (see Section 4.1) to inspect the top SAE features attributed to two different induction heads and find one which specializes in long prefix induction (Goldowsky-Dill et al., 2023), while the other primarily does short prefix induction", where qualitative findings are directly stated without providing any details such as how such inferences are being made by the researchers. This renders the method and its results hard to reproduce, and its interpretability benefits impossible to independently verify. The qualitative nature of analysis alone is not a problem, as such analysis can be made systematic and rigorous using well-designed human studies, which, unfortunately, are missing in this paper.

- This paper does not introduce any novel methods, ideas, or insights into interpretability. Rather, it claims that using SAEs on attention layers results in improved interpretability benefits, which, as mentioned above, is completely unverifiable.

- The introduction section in this paper correctly identifies several use cases of mechanistic interpretability, such as (1) identifying and debugging model errors, (2) control and steering of model behavior, and (3) prediction of OOD behavior. Unfortunately, none of these downstream use-cases are demonstrated with the proposed approach; nor are they benchmarked against previous SAE approaches; making it hard to understand the purported advantages of the presented approach.

**Questions:**

N/A

---

> ### Author Response · Authors · 2024-11-22
> **Author response to reviewer kV6P's feedback**
>
> Thank you reviewer kV6P for your thoughtful feedback on our paper. However, we disagree with your analysis and are open to further conversation as we hope we can resolve some confusions between us.
>
> First, we want to address the criticism that “critical drawback of this paper is the non-systematic and ad-hoc nature of its analysis.” It is not ad-hoc to study existing phenomena, and we note that in section 4.2, we explicitly build on top of the findings in Goldowsky-Dill et al. Further, our analysis of long prefix induction heads is thorough and we dispute that these “qualitative findings are directly stated without providing any details such as how such inferences are being made”: we describe clearly the experiments in section 4.2 where we distinguish short and long prefix heads through computing an induction score. We also find relevant examples of long prefix induction in the training data and show how corrupting them breaks the long induction mechanism. Finally, we then present clearly how Figure 3a and 3b show that (a) the long prefix head 5.1 fails to perform induction on short prefixes compared to the short prefix head 5.5, and that (b) head 5.5 continues to show an induction attention pattern even after intervening on examples that break the induction for head 5.1. Finally, we will open source our weights and experiments upon successful publication, addressing the claim that our work is “unverifiable”.
>
> # Addressing additional concerns
>
> > For example, the paper includes phrases such as "we manually inspect these features for all 144 attention heads in GPT-2 Small. Broadly, we observe that features become more abstract in middle-layer heads and then taper off in abstraction at late layers", and "We use our weight-based head attribution technique (see Section 4.1) to inspect the top SAE features attributed to two different induction heads and find one which specializes in long prefix induction (Goldowsky-Dill et al., 2023), while the other primarily does short prefix induction", where qualitative findings are directly stated without providing any details such as how such inferences are being made by the researchers.
>
> We wanted to apologize about a formatting error in Appendix S.3, which provides a systematic overview of our findings for each attention head. We have fixed this error and have updated the camera ready to omit 4.1, instead placing it in Appendix I, in order to make space in the main body for the other results.
>
> > This renders the method and its results hard to reproduce, and its interpretability benefits impossible to independently verify. The qualitative nature of analysis alone is not a problem, as such analysis can be made systematic and rigorous using well-designed human studies, which, unfortunately, are missing in this paper.
>
> We will open source our weights and experiments upon successful publication to ensure that these findings can be independently verified. We also produce a full set feature dashboards for the systematic analysis of all attention heads in Appendix I.
>
> > The introduction section in this paper correctly identifies several use cases of mechanistic interpretability, such as (1) identifying and debugging model errors, (2) control and steering of model behavior, and (3) prediction of OOD behavior. Unfortunately, none of these downstream use-cases are demonstrated with the proposed approach;
>
> We agree that we did not execute these use cases in our paper. However, we note that the purpose of our paper is to introduce a new methodology through Attention Output SAEs, rather than produce applications of our technique – our introduction is just explaining why people do mechanistic interpretability research in general. Due to the space consumed by multiple extensive case studies, we leave these sorts of use cases for future work rather than add to an already large compilation of interpretability insights produced through Attention Output SAEs in the current paper.
>
> > nor are they benchmarked against previous SAE approaches; making it hard to understand the purported advantages of the presented approach.
>
> This is the first application of the seminal work of Bricken et al [1] to attention SAEs, and there are no good reference points of comparison. We provide an evaluation of the sparsity and reconstruction fidelity of our SAEs in Table 1 for the purpose of allowing downstream users to determine whether the SAEs are high enough quality for their use case, or to allow comparisons to their future work.

---

### Official Review · Reviewer_K6tG · 2024-11-04

**Soundness:** 3
**Presentation:** 3
**Contribution:** 3
**Rating:** 6
**Confidence:** 3

**Summary:**

This paper proposes using sparse autoencoders (SAEs) to mechanistically interpret the outputs of attention layers, similar to SAEs that have previously been used to understand LLM residual streams and MLP layers. The contributions of this work include open-sourcing many trained SAEs, qualitative analysis of the types of features learned by these SAEs, and providing a case study of using these attention layer SAEs to better understand the Indirect Object Identification circuit.

**Strengths:**

- The paper is clear, the writing is easy-to-understand, and the experiments are sound and logical.
- The IOI case study presents an interesting causal analysis of the findings that backs up what are otherwise mainly qualitative results.
- The insights into induction heads and the difference between “long” and “short” prefix induction are novel and interesting.

**Weaknesses:**

- Do the authors believe that the insights from this paper are specific only to attention output SAEs, or that the same insights could be derived from MLP or residual stream SAEs of great enough size? Further discussion of why the results are specific to attention output SAEs would significantly strengthen the contribution of the paper.
- One of the contributions of this work is finding that SAEs learn /interpretable/ features when applied to attention outputs; however, it seems that interpretability is not well-defined in this work and is measured quite qualitatively/ad-hoc, as acknowledged by the authors in the limitations. As such, it would be very helpful if some random examples of interpretable and uninterpretable features were provided in the main paper and the process by which they were interpreted, such that reviewers/readers could calibrate their understanding of what it means without having to search through the visualizer.
- Is labeling scalable for attention output SAEs via methods such as auto-interpretability as it is with other SAEs? Without automatic labeling, do the authors see this method as a feasible way to achieve interpretability of language models or attention?

**Questions:**

- Given that prior work notes that looking only at top-activating features when interpreting SAE features is not optimal, is there a reason why this interpretation procedure was chosen (for example, see https://blog.eleuther.ai/autointerp/). Results from labeling with a distribution of activations would be interesting to see.
- What do the error bars in Fig 4 correspond to? It is unclear to me if the results shown are significant given the size of the error/std/etc bars.

---

> ### Author Response · Authors · 2024-11-22
> **Author response to reviewer K6tG's feedback**
>
> Thank you reviewer K6tG for your thoughtful feedback on our paper. We have carefully considered your comments and have made changes to address the concerns that you raised.
>
> First, we address your concern that “one of the contributions of this work is finding that SAEs learn /interpretable/ features when applied to attention outputs; however, it seems that interpretability is not well-defined in this work and is measured quite qualitatively/ad-hoc, as acknowledged by the authors in the limitations.” We note that **in mechanistic interpretability, it is common practice to look for “patterns” in activations (eg see: Cunningham et al. [4]). While we agree that rigorous definitions are ideal, it is not possible to rigorously define every term in language model interpretability at this time**, and as our work introduces a variant of SAEs, we thought it best to follow established methods in the field, despite their limitations. We expect that any weaknesses due to the limited metrics used may also apply to SAEs more generally. We also address these limitations in Appendix C, as referenced in Section 3.1.
>
> # Concerns about insights related to previous techniques
>
> > Do the authors believe that the insights from this paper are specific only to attention output SAEs, or that the same insights could be derived from MLP or residual stream SAEs of great enough size? Further discussion of why the results are specific to attention output SAEs would significantly strengthen the contribution of the paper.
>
> We emphasize that our Attention Output SAEs are the first to find induction features, due to our use of the DFA technique which enables us to see how information is moved through token positions. As one example, note that we do not believe that induction features are unique to attention SAEs, indeed they are highly likely to appear in residual SAEs as attention layers directly add to the residual stream. However, we expect that they are *computed* in attention layers, which are necessary to compute in-context features, and that Attention Output SAEs and DFA allow us to examine this computation, in a way that residual/MLP SAEs do not allow.
>
> > Given that prior work notes that looking only at top-activating features when interpreting SAE features is not optimal, is there a reason why this interpretation procedure was chosen (for example, see https://blog.eleuther.ai/autointerp/). Results from labeling with a distribution of activations would be interesting to see.
>
> We appreciate the reviewer’s remark that looking only at top-activating features can produce interpretability illusions. We note in Section 3.1 that our feature interpretability methodology considers examination of “activating examples from **randomly sampled activation** ranges, giving a holistic picture of the role of the feature,” and not merely top-activating features. Note that in section 3.3 and appendix G and H, we apply a sensitivity/specificity methodology, wherein we define a proxy for the initial hypothesis interpretation of a feature and then leverage distribution of activations across the entire spectrum of activations to validate interpretation of specific features (cf Figures 2 and 7).
>
> # Addressing remaining points
>
> > Is labeling scalable for attention output SAEs via methods such as auto-interpretability as it is with other SAEs? Without automatic labeling, do the authors see this method as a feasible way to achieve interpretability of language models or attention?
>
> Yes, scalable labeling is feasible. Since the submission of our manuscript, all of our Attention Output SAEs on GPT-2 Small have been automatically labeled through the Neuronpedia project. This is cited in the paper in Appendix A. We provide one example of the “activating on a name previously preceding an ‘and’ token” feature referenced in Section 4.3 (see https://www.neuronpedia.org/gpt2-small/5-att-kk/7515) which was labeled as “proper nouns, particularly names of individuals and notable entities” by gpt-4o-mini. (Note that Neuronpedia is hosting an anonymized presentation of our SAEs to conform to author anonymity requirements.)
>
> > What do the error bars in Fig 4 correspond to? It is unclear to me if the results shown are significant given the size of the error/std/etc bars.
>
> The bars in Figure 4 refer to the min/max ranges of the IOI logit diff for the noising experiment described in the “Confirming the hypothesis” sub-section of 4.3. We note that in such a noising experiment, we need to consider a set of prompts from a similar template to validate the hypothesis, resulting in some variation. However, we note that across most pairs of clean and corrupted prompts, the “rand toks + filler” IOI logit diff exceeded the “alongside” logit diff.

---

### Official Review · Reviewer_pjcH · 2024-11-04

**Soundness:** 2
**Presentation:** 3
**Contribution:** 3
**Rating:** 6
**Confidence:** 3

**Summary:**

This paper applies sparse autoencoders (SAEs), a popular technique in mechanistic interpretability, to the output of attention layers in pre-trained Transformer models, and develops a novel attribution method to deal with multiple attention heads. Using these techniques, the authors arrive at several insights such as identifying specific feature families in GPT-2, identifying different roles of the attention heads such as "long" and "short prefix induction", and a better understanding of position features for "indirection object identification". Based on these findings, the authors make the case for SAEs applied on attention layer outputs as an effective tool for mechanistic interpretability in transformers.

**Strengths:**

**1.** This work makes progress on an important question, namely mechanistic interpretability in transformer models.

**2.**  I found the paper to be very well written. Namely, the writing is concise and clear, and well motivates the authors approach and the problem setting in general.

**3.** I very much appreciated how the authors describe their contribution in the introduction, and avoid making any overclaims. I also found the discussion of limitations throughout the paper to be very transparent.

**4.** The findings presented by the authors, particularly in Section 4.2 and 4.3, I found to be interesting and potentially impactful.

**5**. More generally, the empirical study conducted by the authors appears to be very thorough and varied, with numerous interesting findings. However, I cannot say this with complete confidence as I am not an expert in interpretability.

**Weaknesses:**

**1.** In terms of novelty, the paper is somewhat limited as it is essentially applying existing techniques to a new setting. However, I don't view this as a particularly notable weakness, and importantly, the authors are very upfront about this and their contribution in general.

**2.** One issue I had with this work is that I found the overall structure and presentation of results to be a bit convoluted and unfocused. Specifically, the authors have a large number of findings they want to include and discuss in the manuscript. However, presumably due to space constraints, many of these findings are discussed somewhat superficially in the main text and then the appendix is referenced for further details. Such appendix references occur very often throughout the paper, which made several findings feel rather non-thorough and superficial to me. To this end, I think the paper would have benefitted from focusing more thoroughly on a smaller number of core results.

**3.** My main issue with this work is that I found it difficult to interpret several experimental findings. Specifically, there are several instances throughout the paper, where I felt like the authors presented their findings, without sufficient explanations or experimental evidence to substantiate the claims. I think this was done for reasons similar to those described in **2.**. Namely, that the authors had many findings they wanted to discuss but limited space. Consequently, however, I often felt that I was just taking the authors word for a claim, and was not presented with evidence in the manuscript to understand the validity of the claim.

For example, in Section 4.1, the authors give a description of their findings on the role of different attention heads in GPT-2. However, I did not find any evidence presented to the reader substantiating these claims. Additionally, in Section 3.3 the authors discuss that there are features which take the role of "induction features". However, I found it difficult to understand how the authors verified this based on what was presented in the main text. I presume this was done via Figure 2, however, I had challenges making sense of this empirical protocol based on the explanations given. I am not an expert in this space, so it could be that I am missing something. However, perhaps the authors could explain in a bit more detail, how the "board induction" feature was identified and how this hypothesis was substantiated empirically.

**Questions:**

**1.** The authors mention three different feature families found in 3.3. How did the authors arrive at these three categories. At the surface, they feel a bit arbitrary, so I am curious how they were formulated.

**2.** Can the authors provide some empirical evidence, e.g. a few visual examples, to better understand the findings in 4.1.

**3.** As discussed above, can the authors give a deeper explanation of how the induction features in 3.3 were discovered and the empirical evidence to back this up.

---

> ### Author Response · Authors · 2024-11-22
> **Author response to reviewer pcjH's feedback**
>
> Thank you reviewer pcjH for your thoughtful feedback on our paper. We have carefully considered your comments and have addressed some of your concerns by making appropriate changes in the paper. There were also several items on which we disagreed, and have produced some arguments in response below.
>
> First, to address the criticism that “the authors have a large number of findings they want to include and discuss in the manuscript” and “appendix references occur very often throughout the paper, which made several findings feel rather non-thorough and superficial to me,” we highlight that our main contribution is introducing a new technique, Attention Output SAEs, and also providing a broad evidence base to justify the use of this technique in interpretability research. We note that **this is the first sparse autoencoder paper that simultaneously 1. Introduces a technique, and 2. Applies this technique to open interpretability questions. We thus believe that testing the application of our technique across many aspects of interpretability is an important aspect of the paper. We hope this response convinces you of this claim and look forward to clarifying this further.**
>
> # Responses to Specific Feedback
>
> > in Section 3.3 the authors discuss that there are features which take the role of "induction features". ... However, perhaps the authors could explain in a bit more detail, how the "board induction" feature was identified and how this hypothesis was substantiated empirically.
> > As discussed above, can the authors give a deeper explanation of how the induction features in 3.3 were discovered and the empirical evidence to back this up.
>
> In terms of how the features were discovered, we initially identified the “board induction” feature through inspecting 30 features at random in layer 1 of the Gelu-2L model. We later found 500+ similar instances by automatically detecting such induction features for different tokens, explained in more depth in Appendix I.
>
> In terms of building confidence that this feature was indeed related to “board induction,” we (a) used the feature interpretability methodology in 3.1 to verify that the feature activated on tokens preceding “board” after a previous “X board” pattern at a variety of activation ranges, (b) and (section 3.3) conducted a sensitivity and specificity analysis which demonstrates that at higher activations, the proxy that checks for “board” induction coincides with the feature activation. Note that the specificity and sensitivity analysis is standard as it builds on the work from Bricken et al [1].
>
> > For example, in Section 4.1, the authors give a description of their findings on the role of different attention heads in GPT-2. However, I did not find any evidence presented to the reader substantiating these claims.
> > Can the authors provide some empirical evidence, e.g. a few visual examples, to better understand the findings in 4.1.
>
> We note that the results in Section 4.1 are based on analyzing feature dashboards which is a standard technique in the interpretability field for performing broad stroke analyses of the interpretability of features, for example as in [2] [3]. However, to address the reviewer’s concern, we have moved section 4.1 to Appendix I and refocused section 4 on the concreteness of the results in formerly sections 4.2 and 4.3 (now sections 4.1 and 4.2). We have also added additional figures in Appendix I to provide clear examples of each category of role.
>
> > The authors mention three different feature families found in 3.3. How did the authors arrive at these three categories. At the surface, they feel a bit arbitrary, so I am curious how they were formulated.
>
> We emphasize that we do not make any claims about language model features being cleanly separated into three feature families. Rather, we are organizing our findings into a set of discoveries to show multiple lines of evidence that Attention Output SAEs provide useful insights into models, rather than purely studying single case studies. While features in these feature families were common than those uncategorized, we note that in Appendix P we also find features corresponding to behaviors studied in prior work (succession, name moving, suppression, etc).
>
> > One issue I had with this work is that I found the overall structure and presentation of results to be a bit convoluted and unfocused. [etc]
>
> We appreciate the reviewer’s remark around the overall structure and presentation of the results. As noted by the reviewer, the large volume of results made it challenging to include everything in the main body due to space constraints. As one improvement, we have moved former section 4.1 on interpreting all attention heads in GPT-2 Small to Appendix I in order to refocus the important section 4 on results that demonstrate key insights in interpretability obtained through Attention Sparse Autoencoders.
>
> (For references, see main response.)

---

> ### Author Response · Authors · 2024-11-25
>
> Before this phase of the rebuttal period ends, we wanted to ask the reviewer whether we have addressed your concerns with our work?

---

> ### Author Response · Authors · 2024-12-02
> **Have we addressed your concerns?**
>
> Hello reviewer pjcH, **we only have 23 hours left to be able to respond to your questions** and concerns. We think that we've addressed them as detailed in our rebuttal to you, so could you let us know ASAP about further worries, or otherwise would you be open to revising your score?

---

> > ### Comment · Reviewer_pjcH · 2024-12-02
> >
> > I believe my concerns stated on Nov. 26 regarding the overall presentation structure and ad hoc nature of many findings still stand, and, thus, I will refer the authors to this response, which was unanswered.
> >
> > In general, I think these concerns deal with more fundamental issues I have with this work, opposed to minor issues which can be quickly fixed. With this being said, I believe I have been fair to the authors in recognizing the potential merit of their work despite these concerns. I stand by this assessment and will thus stick with my score and recommendation.

---

### Meta-Review · Area_Chair_gyCt · 2024-12-20

**Metareview:**

The paper introduces a method to apply sparse autoencoders to interpret attention layer outputs in transformers.

Despite some reviewers acknowledging the potential interest for the interpretability community, the main reservations remained: the ad hoc interpretability metrics, the lack of solid comparative benchmarks, and insufficient demonstration of how this method uniquely advances the field. While the concept of attention output SAEs is intriguing, the reviewers did not feel the paper fully established its rigor and practical significance.

Given these considerations, for the benifits of this paper, we recommand rejecting this paper for now. Note that this is not a disencouragement. We believe that this paper can be a strong submission after addressing the remaining conerns.

**Additional Comments On Reviewer Discussion:**

During the rebuttal and discussion phase, the reviewers appreciated the paper’s contributions to the mechanistic interpretability of large language models. However, during Reviewer-AC discussion, reviewers acknowledge that some major concerns remained.


Reviewers consistently flagged issues with the qualitative and ad hoc nature of the interpretability claims, noting the need for more systematic and reproducible validation methods. Specifically,

   - *Reviewer K6tG* stated that “the authors’ notion of interpretability is not very well-defined and could benefit from more transparency or potentially a small user study with reported confidence intervals.” This highlights the need for clearer metrics or systematic validation to substantiate interpretability claims.

   - *Reviewer pjcH* acknowledged the breadth of results but noted that “several findings presented in this work have an ad hoc nature to them, which makes it difficult to have complete confidence in the findings.”

   - *Reviewer kV6P* emphasized reproducibility challenges, pointing out that “central parts of the analysis are qualitative and performed by the authors themselves,” and warned about the potential for confirmation bias in such subjective evaluations. The Reviewer argued that “rigorous user studies are essential for qualitative experiments involving human judgments, especially around questions of interpretability.”


Due to the foundational nature of the concerns, major revisions should be required to address the issues raised. While the work has potential, it would benefit from further refinement before acceptance.

---

### Decision · Program_Chairs · 2025-01-22

Reject